# In situ observation of crystal rotation in Ni-based superalloy during additive manufacturing process

Dongsheng Zhang [1,2,6], Wei Liu [3,6], Yuxiao Li [4], Darui Sun [1], Yu Wu [3], Shengnian Luo [4], Sen Chen [5] ✉, Ye Tao [1] ✉ & Bingbing Zhang [1,2] ✉

Understanding the dynamic process of epitaxial microstructure forming in laser additive manufacturing is very important for achieving products with a single crystalline texture. Here, we perform in situ, real-time synchrotron Laue diffraction experiments to capture the microstructural evolution of nickel-based single-crystal superalloys during the rapid laser remelting process. In situ synchrotron radiation Laue diffraction characterises the crystal rotation behaviour and stray grain formation process. With a complementary thermomechanical coupled finite element simulation and molecular dynamics simulation, we identify that the crystal rotation is governed by the localised heating/cooling heterogeneity-induced deformation gradient and recognise that the sub-grain rotation caused by rapid dislocation movement could be the origin of granular stray grains at the bottom of the melt pool.

Additive manufacturing (AM), such as selective laser melting (SLM) and direct energy deposition (DED), is driven by a three–dimensional (3D) digital model and used in the direct manufacturing of complex structures without any moulds; moreover, this process has been considered a revolutionary breakthrough in the field of manufacturing technology[1–3]. During the manufacturing process, the metal powder is melted instantly, and a microscale molten pool forms in a short time under the operation of a high-energy laser or electron beam, achieving steep temperature gradients of up to ~$10^7$ K m$^{-1}$ and ultrahigh cooling rates of ~$10^7$ K s$^{-14}$. As a consequence, the prominent microstructural feature of printed metallic materials is columnar grains that grow epitaxially from the melt pool boundary[5–8], which is often undesirable for industrial applications[9,10]. However, for nickel-based single-crystal (SX) superalloys used in high-temperature environments, using the epitaxial growth characteristics of AM to obtain directional or single-phase alloys is essential.

There have been numerous investigations on the epitaxial growth of nickel-based SX superalloys during AM processes. Santos et al.

demonstrated[11] that the deposition of René N4 nickel superalloy grows epitaxially and inherits the orientation of the substrate, presenting a columnar dendritic structure with <100> orientation during laser melting deposition (LMD). Liu et al. developed[12] a self-consistent 3D mathematical model to predict the crystal growth and microstructure formation in the LMD of SX superalloys. Liang et al. investigated[13] the influence of the processing parameters on the deposited productivity and epitaxial SX growth in the LMD process by the orthogonal experiment method. Chauvet[14] and Lin[15] et al. controlled the selective electron beam melting process window and successfully fabricated SX superalloys on stainless steel substrates after growth competition of different grains in the first few layers without a separator and seed crystals. Very recently, SX nickel was manufactured by a flat-top laser[16]. However, previous studies are mainly based on the characterisation of the recovery specimens and numerical simulations. The direct characterisation of dynamic behaviours in the melt pool is very challenging and remains unknown, hindering a deep understanding of the rapid melting and solidification behaviour of SX superalloys.

[1]Institute of High Energy Physics, Chinese Academy of Sciences, 19B Yuquan Road, Beijing 100049, P R China. [2]University of Chinese Academy of Sciences, 19A Yuquan Road, Beijing 100049, P R China. [3]3D Printing Research & Engineering Technology Center, AECC Beijing Institute of Aeronautical Materials, Beijing 100095, P R China. [4]The Peac Institute of Multiscale Sciences, Chengdu, Sichuan 610207, P R China. [5]Institute of Fluid Physics, China Academy of Engineering Physics, Mianyang, Sichuan 621900, China. [6]These authors contributed equally: Dongsheng Zhang, Wei Liu. ✉e-mail: chensen@mail.ust.edu.cn; taoy@ihep.ac.cn; zhangbb@ihep.ac.cn

With high temporal and spatial resolutions, synchrotron radiation-based ultrafast X-ray techniques enable the monitoring of microstructural dynamics during the AM process in situ and in real time[17,18]. Zhao et al. first demonstrated[19] the great potential of ultrafast X-ray imaging and diffraction in characterising microscale defects and structures during the AM process. Since then, many works involving ultrafast X-ray imaging to capture microdefect motions of key holes, pores and hot cracks have emerged and achieved great results[20–24], but far fewer works have appeared using X-ray diffraction to monitor the microstructural dynamics in the melt pool. Time-resolved X-ray powder diffraction has been utilised to reproduce the fast α to β phase transitions of Ti6Al4V during the AM process[25–27] as exemplary experiments but has limitations in characterising single crystal specimens, such as SX superalloys. Some efforts are yet missing to capture the rapid melting and solidification process of SX superalloys, as is the mechanistic research on epitaxial growth behaviour during the remelting process.

In this work, we conduct in situ, real-time X-ray Laue diffraction experiments to study the microstructural evolution of the second-generation nickel-based SX superalloy during the laser remelting process, capturing the dynamic crystal rotation behaviour and stray grain (SG) formation in the melt pool. In combination with the thermomechanical coupled infinite element method and molecular dynamics simulation, we reveal that the localised heating heterogeneity-induced material deformation gradient governs crystal rotation. Moreover, the potential origin of granular stray grains at the bottom of the melt pool is also discussed. The in-depth understanding of the crystal rotation and SG formation mechanisms could be helpful for optimising additive manufacturing approaches to print perfect products with single-crystal texture.

## Results

### Experimental setup and ex-situ microstructure characterisation

The experimental setup consists of a bare plate (SX nickel-based superalloy with 0.8 mm thickness), a selective laser melting system and an in situ Laue diffraction experimental system, as shown in Fig. 1a (see "Methods" section for details). The chemical composition of the SX nickel-based superalloy is given in Supplementary Table 1. Laue diffraction was conducted with a frame rate of 200 Hz. The laser power ($P$), laser scanning speed ($v$), laser beam diameter ($d$) and laser scanning length with a single track used in this experiment were 266 W, 0.02 m s$^{-1}$, 800 μm and 2 mm, respectively, corresponding to an input energy density (IED)[20] of 1662.5 J cm$^{-2}$ (IED = $P/(v \cdot d)$). The 100 μm (full width at half height, FWHM) X-ray beam spot was located in the middle of the laser scanning path and 200 μm below the top surface of the sample.

To characterise the initial and final states of the substrate, optical microscopy (OM) and electron backscattered diffraction (EBSD) of the remelted sample were conducted. As shown in Fig. 1b, the OM image shows the typical feature of epitaxial growth of columnar grains originating from the melt pool boundary. The newly manufactured columnar dendrites are smaller, and many sub-grain boundaries appear. Equiaxed grains are observed at the top of the melt pool where the columnar to equiaxial transition usually occurs[28,29]. Figure 1c reveals the crystallographic orientations in the **Z** direction (or building direction). As shown in Fig. 1 (c1), the γ phase of the as-cast region of the substrate mainly exhibits one crystallographic orientation in the **Z** direction, which is significantly different from that after laser melting (Fig. 1(c2)). An obvious crystallographic orientation shift can be observed along the growth direction of dendrites, indicating a crystal rotation of ~1.9° between the initial and final state along the **Y** axis (Y-rotation). Hence, to further reveal the "missing" crystal rotation behaviours during the laser remelting process, it is necessary to carry out in situ, real-time characterisation.

### In situ Laue diffraction during the laser remelting process

The microstructural dynamics of the sample during the laser remelting process were temporally tracked by in situ Laue diffraction measurements. Figure 2 shows the representative temporal sequence of Laue diffraction patterns during the laser remelting process. Complete details can be found in Supplementary Movie 1. Eleven high-intensity diffraction spots of the substrate were captured (Fig. 2a) and successfully indexed (Supplementary Fig. 1). These diffraction spots split into two subpeaks (Fig. 2f) before laser melting, indicating the existence of two orientation-separated sub-grains, i.e., the geometrically necessary boundaries in the X-ray probed region (XPR)[30,31]. The diffraction image of Fig. 2b captured at 225 ms shows a broad, liquid-phase scattering ring, referring to the complete melting moment of the SX substrate within the XPR.

Figure 2c indicates the critical start point of solidification. It was found that the orientation of the dendrites was similar to that of the substrate, indicating that the epitaxial dendrites essentially inherited the substrate orientation. Figure 2d shows the termination of epitaxial growth within the XPR; afterwards, the intensity of the Laue spots hardly increases. Sporadic diffraction spots, originating from the SGs whose orientations were different from the epitaxial grains, also appeared. The low-intensity Laue spots of SGs imply a small grain size. Figure 2d, e refers to the cooling process, where little change in the number of Laue spots and a slight increase in the intensity are exhibited, indicating the generation of few SGs. Therefore, we can deduce that it is during solidification that the epitaxial-to-stray grain transition appears. Figure 2f shows the enlarged Laue spots before and after

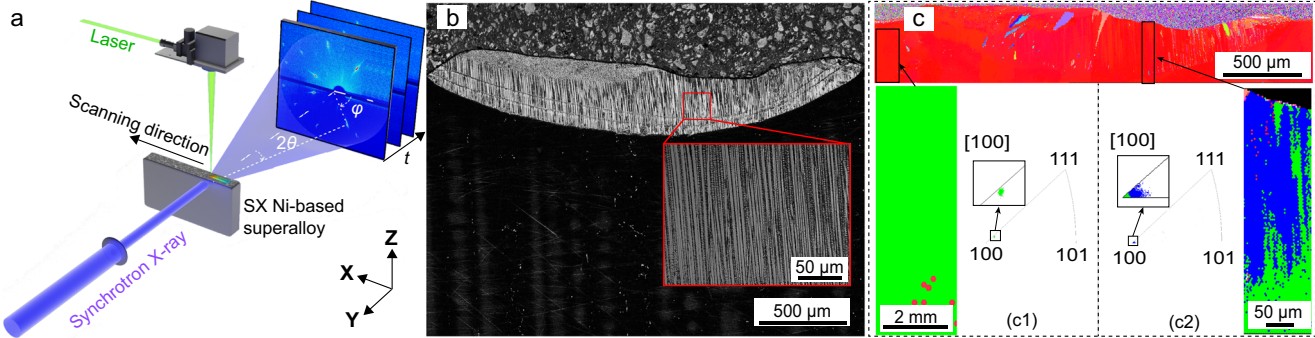

**Fig. 1 | Characterisation of the microstructural evolution of nickel-based SX superalloy during the laser remelting process. a** Schematic rendering of the in situ laser remelting setup, along with the coordinate system (**XYZ**). **b** The OM image of the remelted plate. **c** EBSD results of the as-cast region (c1) and the after-remelted region (c2), where the [100] direction refers to the **Z** direction. In (c2), the blue and green colors indicate the (1 0 30) and (0 0 1) crystal planes along the **Z** direction, respectively.

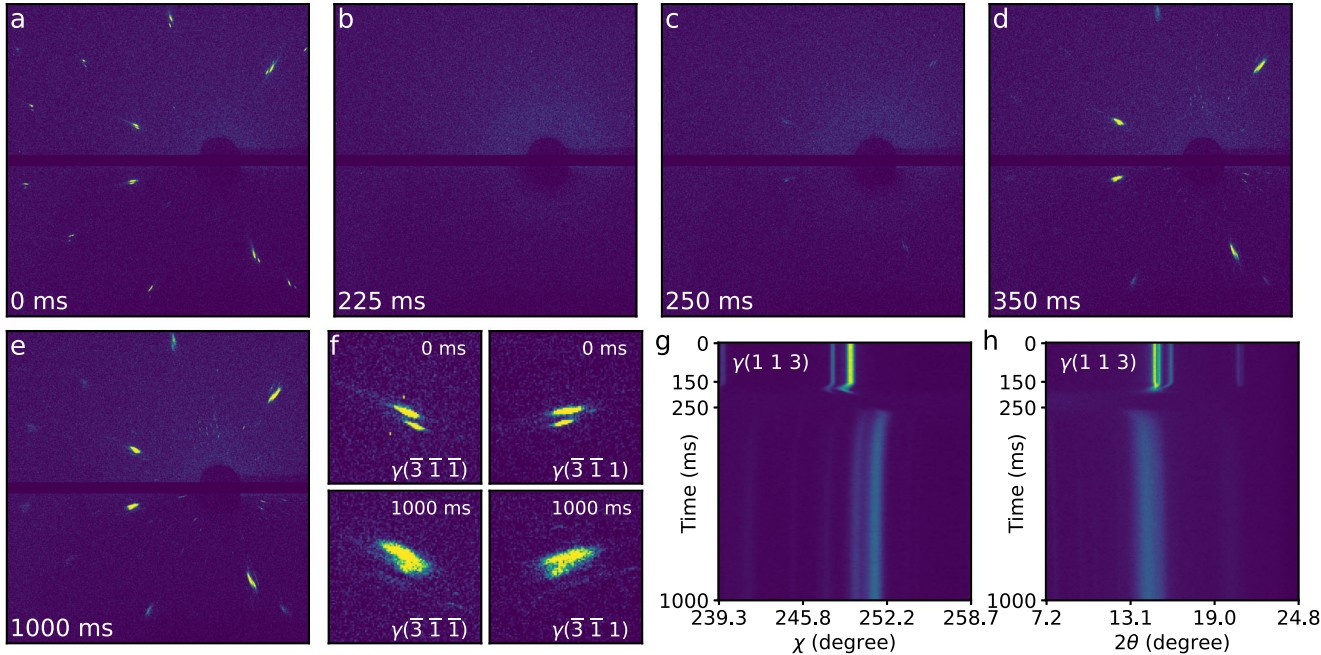

**Fig. 2 | Time series of the representative Laue diffraction patterns during laser remelting processes. a–e** Laue diffraction images collected at 0 ms (**a**), 225 ms (**b**), 250 ms (**c**), 350 ms (**d**) and 1000 ms (**e**). **f** Local enlarged drawing of the diffraction spots of γ($\bar{3}1\bar{1}$) and γ($\bar{3}11$) lattice planes at 0 and 1000 ms. **g, h** Variation diagrams of the γ(1 1 3) crystal plane with time in the χ direction (**g**) and 2θ direction (**h**). The laser was switched on at 150 ms and off at 250 ms.

remelting, where the splitting of the diffraction spots still remains, indicating that geometrically necessary boundaries cannot be eliminated by laser remelting.

The temporal evolutions of the γ(113) diffraction peak in the χ and 2θ directions are displayed in Fig. 2g, h, respectively. Prior to the full melting of the XPR (150–200 ms), the diffraction peak apparently shifts. During 200–250 ms, the diffraction peak disappears due to the solid-to-liquid transition. The diffraction peak reappears at 250 ms, accompanied by peak broadening and peak shifting.

To quantitatively extract the temporal evolution of the diffraction angular shift and FWHM change, four diffraction spots, including γ($\bar{3}1\bar{1}$), γ($\bar{3}11$), γ(1 1 3) and γ(11$\bar{3}$) lattice planes of 200 frames, were fitted by the 2D Gaussian function. Interesting "crystal rocking" behaviour can be found during the remelting process along the χ direction (Fig. 3a). The four lattice planes first rotate anti-clockwise along the **Y** axis to -0.5 degrees in Laser Heating Period I (150–175 ms) and then rotate clockwise in Period II (175–200 ms). In Period III (200–250 ms), the detected region was completely melted, and no Bragg peak could be found. The melt pool begins to solidify while the dendrites start to grow at 250 ms, since the scan laser is ceased at 250 ms. During the solidification period (250–350 ms), the rotation angle, i.e., the angular deviation between the epitaxial growth direction of the primary dendrite and the initial substrate, first shows a sharp rise to -2.5° and then a very slow decay. This deviation further decreases to 1.5° ~ 1.9° in the following cooling process (350–600 ms) due to the anti-clockwise crystal **Y**-rotation.

Figure 3c shows the rotation angle (in the 2θ direction) of the four lattice planes versus time. Although both crystal rotation along the **X**-axis and **Z**-axis can introduce changes in 2θ, the difference between **X**- and **Z**-rotations lies in the shifting directions of the diffraction peaks. If the crystal rotates along the **X**-axis, the 2θ change in the γ($\bar{3}1\bar{1}$) and γ(11$\bar{3}$) lattice planes would be opposite to that of γ($\bar{3}11$) and γ(113). This result is consistent with Fig. 3c, indicating that the crystal mainly rotated along the **X**-axis.

Figure 3b, d depict the FWHM change of the four lattice planes in the χ and 2θ directions versus time, respectively. During Laser Heating Periods I and II, the FWHM increases in both the χ and 2θ directions,

which may be attributed to inhomogeneous strain or lattice bending introduced by local thermal inhomogeneity. At the initial state of the liquid-to-solid transition, the broadening of the diffraction peaks is the largest. As solidification proceeds (red rectangular zone), the FWHM dramatically decreases. The eventual FWHM of epitaxial dendrites is larger than that of the initial substrate, indicating an inhomogeneous dendritic microstructure.

## Influence of the laser power on stray grain content

The Laue diffraction spot number for a single crystal material can reflect the SG behaviours to some extent, because the SG orientation is different from that of the epitaxial grains. The diffraction spot number as a function of time is illustrated in Fig. 4a, where we can obtain the evolution of the diffraction spots during the melting and solidification processes at a laser power of 266 W. During the laser heating periods, the number of diffraction spots decreases, indicating solid-to-liquid phase transformation. After the laser was switched off, the sample began to solidify. As indicated by the red rectangular zones of Fig. 4a, the number of spots rapidly increases, after which there are few changes in the number of Laue spots. Thus, we can deduce that SGs appear during the solidification process.

To investigate the influence of the laser power on the content of SGs, we calculated the number of diffraction spots after laser melting as a function of the laser power. With a higher content of SGs, more sporadic diffraction spots appear. Figure 4b illustrates clear proportional relationships of the diffraction spot number with respect to the laser power, indicating that the content of SGs within the remelted region significantly increases with respect to the laser power. Note that the diffraction spot number of the substrate before the laser remelting is the same as that after 245-W laser remelting, indicating that few SGs appear in the XPR under an appropriate power of 245 W. The original Laue diffraction image series, as well as the spot number calculation method can be found in the supplementary material (Supplementary Movies 2–4 and Supplementary Fig. 3).

Figure 4c also shows the EBSD images of the remelted substrates at different laser powers, where we can find that there are small granular SGs located at the boundary of the melt pool and that

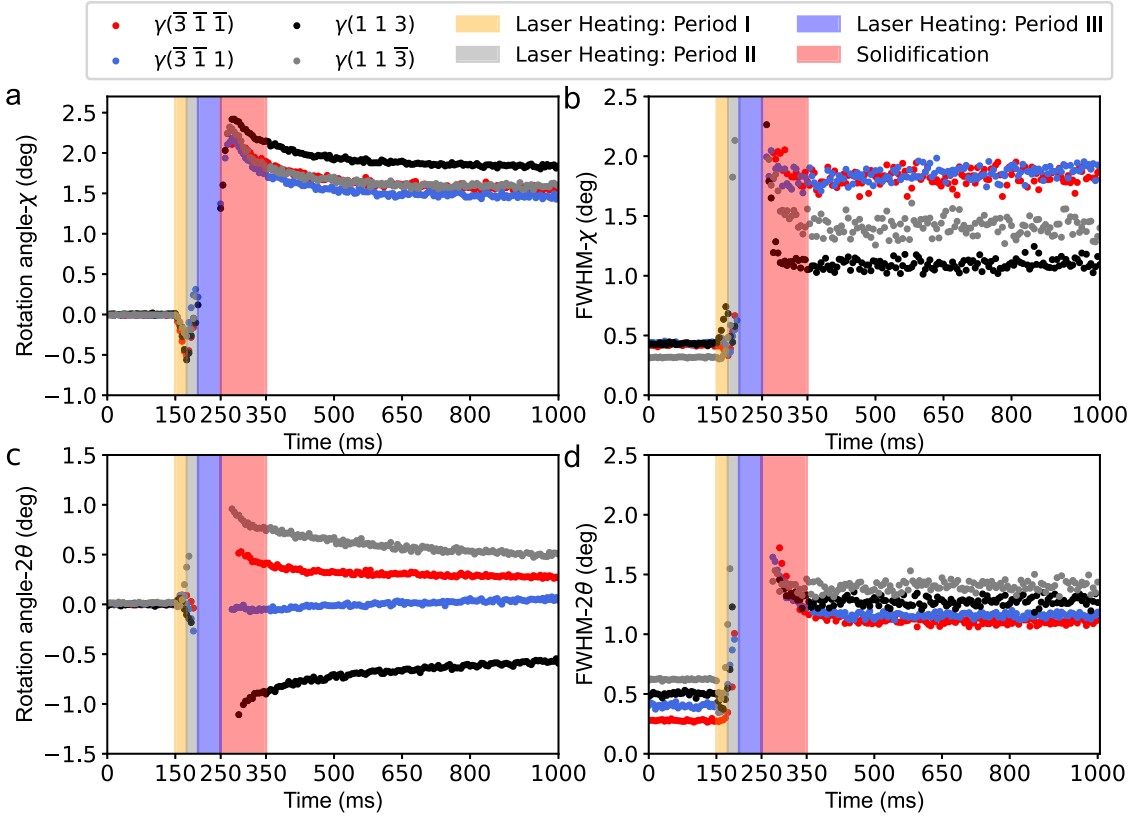

**Fig. 3 | Dynamic evolution of the diffraction peaks during the laser remelting process. a**, **b** The rotation angle (**a**) and FWHM (**b**) in the $\chi$ direction versus time. **c**, **d** The rotation angle (**c**) and FWHM (**d**) in the $2\theta$ direction versus time. The red, blue, black and grey dots represent the $\gamma(3\bar{1}\bar{1})$, $\gamma(3\bar{1}1)$, $\gamma(113)$ and $\gamma(11\bar{3})$ lattice planes, respectively. The orange, grey, purple and red rectangular zones denote four different stages of the laser remelting process. The laser was switched on at 150 ms and off at 250 ms.

equiaxed SGs more easily exist at the surface of the melt pool. Moreover, the content and grain size of SGs within the remelted region significantly increases with respect to the laser power. The consistency between the in situ Laue diffraction and the ex situ EBSD results paves the way for the highly efficient optimisation of the additive manufacturing process.

It is important to note that the same in situ Laue diffraction experiments were performed on both a bare plate and with powders (Supplementary Movie 5). Similar crystal rotation behaviour and the SG content-laser power dependency were also observed, as shown in Supplementary Figs. 4 and 5.

## Discussion

The rotation of the crystal orientation during deposition hinders the printing of single crystals[32]. Thus, it is important to understand the crystal rotation mechanism. The angular shift of the Laue diffraction spots in Fig. 3 could be caused by both the crystal rotation and the lattice strain. To simplify the analysis model, we first assume that the angular shifts were independently introduced by the crystal rotation and simulate the Laue spot evolutions[33,34] when the crystal rotates along the three axes, as shown in Supplementary Movies 6–8. We can see that the **Y**-rotation mainly corresponds to the angular shift along the $\chi$ direction, while the **X**- and **Z**-rotation could contribute to the shift along the $2\theta$ direction. Further comparison with the experimental data gives the relative coordinates of the Euler angle of the grain orientation at different moments during the laser remelting process, as shown in Table 1. The calculation details are shown in the "Methods" section.

The consistency between $\varphi_y$ and the experimental data indicates the dominant rotation contribution in revealing the origin of the angular shift along the $\chi$ direction. This could be further verified by the

quantification of the deformation effect given by the thermo-mechanical simulation and Laue diffraction simulation, as shown in Supplementary Fig. 2. This figure shows that the strain effect only plays a minor part ($\Delta\chi = 0.15$–$0.4°$) in inducing the angular shift along the $\chi$ direction.

However, the inconsistency between $\varphi_x$, which is obtained under the assumption of crystal rotation, and the experimental angular shift along the $2\theta$ direction indicates a more complex mechanism of coupling multiple factors. The thermomechanical-assisted Laue diffraction simulation results in Supplementary Fig. 2 further verified this outcome, where the strain-induced angular shift indeed plays an important role that cannot be ignored ($0.15 \sim 0.4°$). Considering the relatively small scale of the peak shift along the $2\theta$ direction, only the origin of the **Y**-rotation is discussed in this work.

According to the theory of solid mechanics, the deformation gradient **F** shows how an infinitesimal line element, d**X**, is mapped to the corresponding deformed line element d**x** by[35]

$$d\mathbf{x} = \frac{\partial \mathbf{x}}{\partial \mathbf{X}} d\mathbf{X} = \mathbf{F} d\mathbf{X}$$

where **F** contains the complete information about the local strain and rotation of the material. It can be decomposed by right polar decomposition into a rotation tensor **R**, which describes the rigid-body rotation, and a symmetric tensor **U** containing all information about the deformation of the material, i.e., **F** = **RU**. In the present study, the temporal evolution of the deformation gradient field is a possible reason for crystal rotation and deformation during the laser heating process. The strain tensor $\boldsymbol{\varepsilon}$ can be calculated by **U**. Since **U** is independent of rigid-body rotations, this should also apply to the

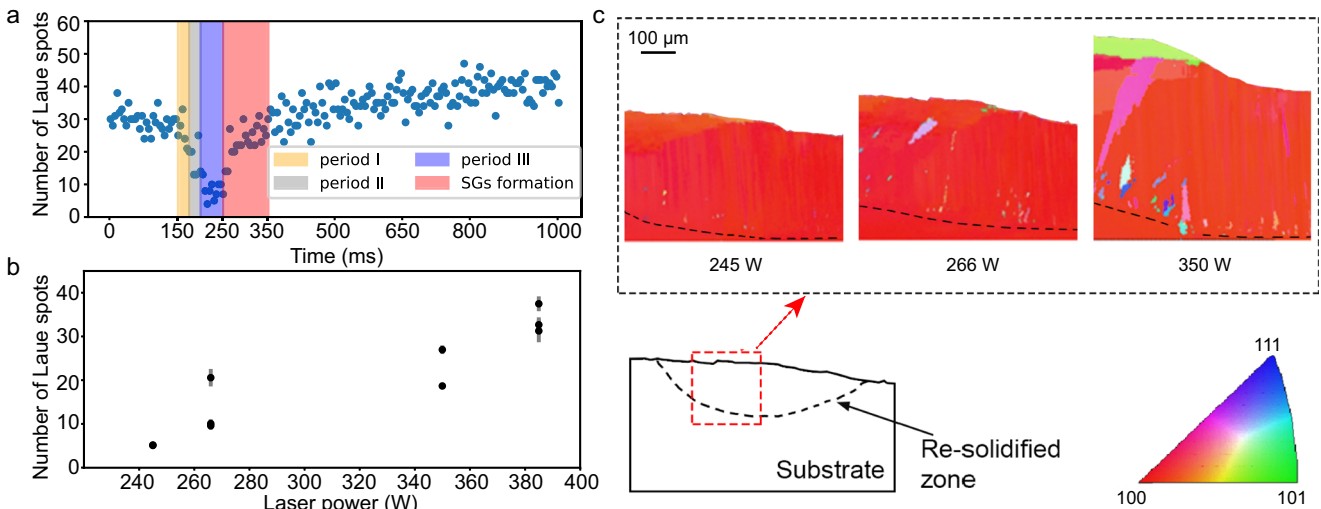

**Fig. 4 | Effect of laser power on the formation of SGs. a** Number of diffraction spots as a function of time at a laser power of cm W. The orange, grey, purple and red rectangular zones denote Laser Heating Period I, Laser Heating Period II, Laser Heating Period III and solidification and SG formation period, respectively. **b** Number of diffraction spots as a function of the laser power. Error bars indicate the standard deviation coming from 20 repeatedly recorded data in the same sample. **c** Ex situ EBSD images at different laser powers.

strain. As a result, the in situ experimental crystal rotation and deformation should be controlled by the rotation tensor and strain tensor, respectively. To verify this hypothesis, a multi-physics finite element simulation was performed with the same laser processing parameters as the in situ measurements (see "Methods" section for details).

The simulated temperature evolution during the laser remelting process is shown in Fig. 5a, where Laser Heating Period III (200–250 ms) represents the moment when the sample melts. The flat curve at the end of Laser Heating Period III is caused by the latent heat effect, indicating the liquid-solid transition. The mean heating rate is ~2.6 × 10⁴ K s⁻¹. The mean cooling rates during 250–400 ms and 400–1000 ms are ~3.4 × 10³ K s⁻¹ and ~0.8 × 10³ K s⁻¹, respectively.

The temporal evolutions of $\varepsilon_{XX}$, $\varepsilon_{YY}$ and $\varepsilon_{ZZ}$ are shown in Fig. 5b. The representative spatial distributions of the strain field before and after laser scans across the XPR are displayed in Fig. 5c–h, respectively. The XPR is compressed along the **X** and **Z** directions[27] but stretched along the **Y** direction, as the laser scans close the XPR in the first several tens of milliseconds. As the melt pool approaches the XPR, the compression $\varepsilon_{XX}$ and $\varepsilon_{ZZ}$ fields are gradually transformed into tensile fields, thus inducing thermal expansion. When the laser stops at $t = 250$ ms, the melt pool begins to solidify, and the strain gradually decreases.

The temporal evolution of **R** is plotted in Supplementary Fig. 6. **R** is similar to a single **Y**-rotation operation; thus, the crystal **Y**-rotation angle $\chi$ mainly refers to the xZ component of rotation tensor $R_{xZ}$ (or $R_{zX}$), i.e., $R_{xZ} = \sin\chi$. Hence, the **Y**-rotation angle during the laser heating process could be calculated and depicted in Fig. 6. As a consequence of the localised heating heterogeneity, the spatial distribution of the **Y**-rotation exhibits typical polarisation characteristics, with negative rotation at the front end of the melt pool and positive rotation at the back end, as depicted in Fig. 6a–c. Accompanied by laser scanning, the polarised rotation field propagates sequentially through the negative region (Fig. 6a), the critical point (Fig. 6b), where $R_{xZ}$ is smallest and then the positive part (Fig. 6c), resulting in "crystal rocking" in Fig. 6d and Fig. 6e during the melting process.

The experimental and simulated **Y**-rotation angles are plotted in Fig. 6d, e, respectively, where the simulation results clearly reproduce the inversion of the crystal rotation of the experimental data, indicating that the laser-induced temporal evolution of the deformation gradient is the dominant mechanism of the crystal rotation.

In addition, **R** is similar to a single **Y**-rotation operation during the laser heating process, indicating that the **X**- and **Z**-rotations of the crystals are not prominent compared with that of the **Y**-rotation. This outcome is consistent with the in situ experimental results.

In the subsequent solidification and cooling processes, as illustrated in Fig. 7, crystal rotation also occurred. The deformation gradient field is also presented on the solidified crystal as a consequence of the thermal gradient, resulting in crystal rotation during solidification and cooling. The finite element simulated rotation angles reproduce the experimental data well, as shown in Fig. 7a, b. As a result, the orientation of the crystal at the beginning of solidification depends on the temperature gradient, and the following crystal rotation in the subsequent process is due to the evolution of the deformation gradient field.

In the final state, we also found that the epitaxial dendrites essentially inherited the orientation of the initial matrix, with the lowest energy barrier, instead of other competitive solidification paths, such as homogeneous or stray nucleation[36]. The newly solidified grains and matrix metal present the same crystallographic structure, giving rise to a low-energy homoepitaxial interface[37], which eventually causes the expected orientation inheritance and resultant secondary dendritic arms orienting in the same direction as the substrate. However, subtle orientation deviation due to **Y**-rotation could also be found between the newly solidified crystal at 300 ms and the initial matrix, as illustrated in the OM (Fig. 1b), EBSD (Fig. 1c) and in situ experiments (Fig. 3a).

MD simulation verifies these observed orientation changes. Figure 7c, d show the angular difference of the [001] crystal orientation of the epitaxially grown grains relative to the initial direction, and it can be seen that there is an angular deviation of the [001] crystal orientation at different locations of the melt pool. This difference is consistent with the distribution trend of the temperature field, i.e., the [001] crystal orientation of the growing grains is biased towards the direction with the largest temperature gradient. Hence, the deviation between epitaxial dendrites and the substrate was controlled by the largest temperature gradient. Similar crystal rotations have been reported by several previous studies using ex situ metrologies[32,38].

The generation of SGs is also a hindrance to the printing of single crystals in additive manufacturing processes[14,16]. As depicted in Fig. 4c, there are many granular SGs near the boundary of the melt pool and equiaxed SGs at the surface of the melt pool. The regular distribution

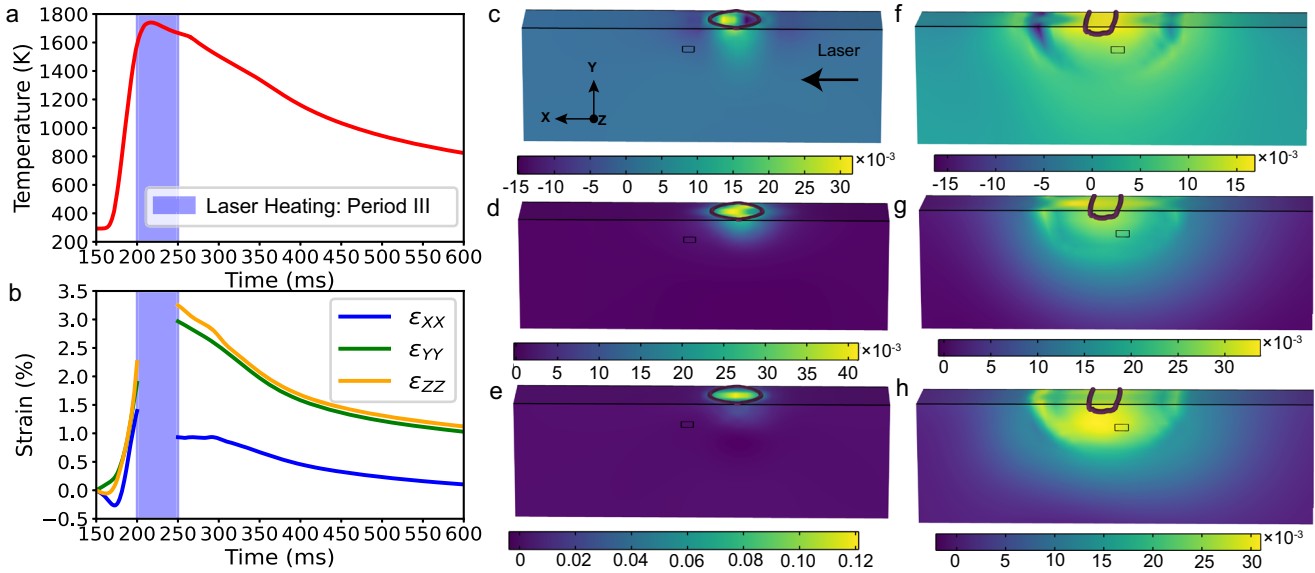

**Fig. 5 | Thermomechanical finite element simulation results.** The evolution of the temperature (**a**) and strain (**b**) of XPR during the laser remelting process. The spatial distributions of $\varepsilon_{XX}$, $\varepsilon_{YY}$ and $\varepsilon_{ZZ}$ at 165 ms and 310 ms are shown in (**c**–**h**), respectively. The brown tubular zones refer to the melt pool boundary, and the cuboids represent the XPR.

of SGs excludes the thermosolutal convective formation mechanism[39] that usually introduces irregularly distributed SGs. Abundant literature on the theoretical and experimental work on the constitutional supercooling mechanism[40–45] explains the dominant mechanism of the formation of equiaxed SGs at the top of the melt pool but cannot explain the formation mechanism of granular SGs near the melt pool boundary. There are studies indicating that defects may play an important role in the formation of SGs[16,28,38]. Hence, we performed the MD simulation to obtain insights into the defect evolution and nucleation of SGs (See Methods section for details). Although the current synchrotron radiation experiments are unable to directly capture the nucleation process of SGs, they do provide significant experimental support for MD simulation.

Typical spatial temperature distributions are shown in Fig. 8a, with exponentially decaying cooling rates. Stress is produced on the solidified crystal due to the temperature gradient effect, resulting in the formation of defects such as dislocations and stacking faults at the boundary of the melt pool. Figure 8b–d shows snapshots of the melt pool with a diameter of 68 nm in single crystal nickel at three different solidification stages (60, 300 and 600 ps, solidification timestep) during the cooling step of 800 ps, where the FCC atoms (green) represent solid crystals, hcp atoms (orange) represent crystal defects such as stacking faults, and the amorphous atoms (white) represent liquid or grain boundary atoms. To identify the local crystalline environment, the polyhedral template matching method (PTM) is used.

At $t = 60$ ps, the solid–liquid interface advances towards the centre of the melt pool, marking the beginning of epitaxial crystal growth,

as shown in Fig. 8b. As the rapid solidification continues, at $t = 300$ ps, new grains are homogeneously nucleated at the centre of the melt pool, as shown in Fig. 8c. At $t = 600$ ps (Fig. 8d), most of the liquid Ni atoms solidified into the crystalline environment. However, the epitaxial growth of grains is thwarted by homogeneously nucleated grains from the centre of the melt pool.

The defect evolution during remelting processing is also shown in Fig. 8b–d. A large number of stacking faults (SFs) form and extend toward the central region of the melting pool. Residual stress is produced on the nickel crystal due to the temperature gradient effect, which can result in the formation of dislocations. The initial partial dislocation gradually forms longer and complete SFs. Evidently, defects are observed at the boundary of the melting pool and grow to the centre.

The temporal evolution of the dislocation density during solidification and its contribution to the FWHM change of the Laue spot are illustrated in Fig. 8e, f, respectively. The diffraction simulation was calculated based on the modified Williamson-Hall method[46–48] with the dislocation density evolution data (Fig. 8e) as the input. A similar FWHM−$2\theta$ change between the simulated (Fig. 8f) and experimental data (Fig. 3d) hints at a direct correlation between the diffraction peak width and dislocation density. Although many factors can contribute to the peak width change of the Laue spots, such as the gradient of the lattice strain, grain size and energy bandwidth, we deconvoluted other factors and found that the dislocation density change plays the most important role in the current work; more discussion is provided in Supplementary Figs. 7–9. Therefore, we attribute the dramatic decrease in the FWHM of Laue spots during the SG formation period to the evolution of the dislocations and obtain the following hypothesis.

At the initial state of solidification, there were dense dislocations around the epitaxial grains, possibly caused by segregated impurity elements[49] or intrinsic localised heating/cooling heterogeneity upon laser scanning[50]. The abundant dislocations acted as preferential nucleation sites, resulting in an increase in the nucleation rate and excessive local constitutional supercooling. Consequently, the material precipitated on the nucleation sites to form fine grains. Meanwhile, complicated and aeolotropic stress fields existed around the grains due to various defects and anisotropic temperature fields. Under the joint action of the rapid moment of dislocation and anisotropic stress field, the fine sub-grains captured many dislocations from epitaxial

**Table. 1 | Euler angle of the grain orientation relative X, Y and Z coordinate during the laser remelting process without considering the influence of lattice strain**

|  | $\varphi_X$ | $\varphi_Y$ | $\varphi_Z$ |
|---|---|---|---|
| Matrix metal at 0 ms (deg) | 0.2 | 5 | −0.5 |
| Matrix metal at 200 ms (deg) | 0.6 | 4.8 | −0.5 |
| Epitaxial dendrite at 300 ms (deg) | 0.8 | 6.5 | −0.5 |
| Epitaxial dendrite at 1000 ms (deg) | 0.6 | 6.9 | −0.5 |

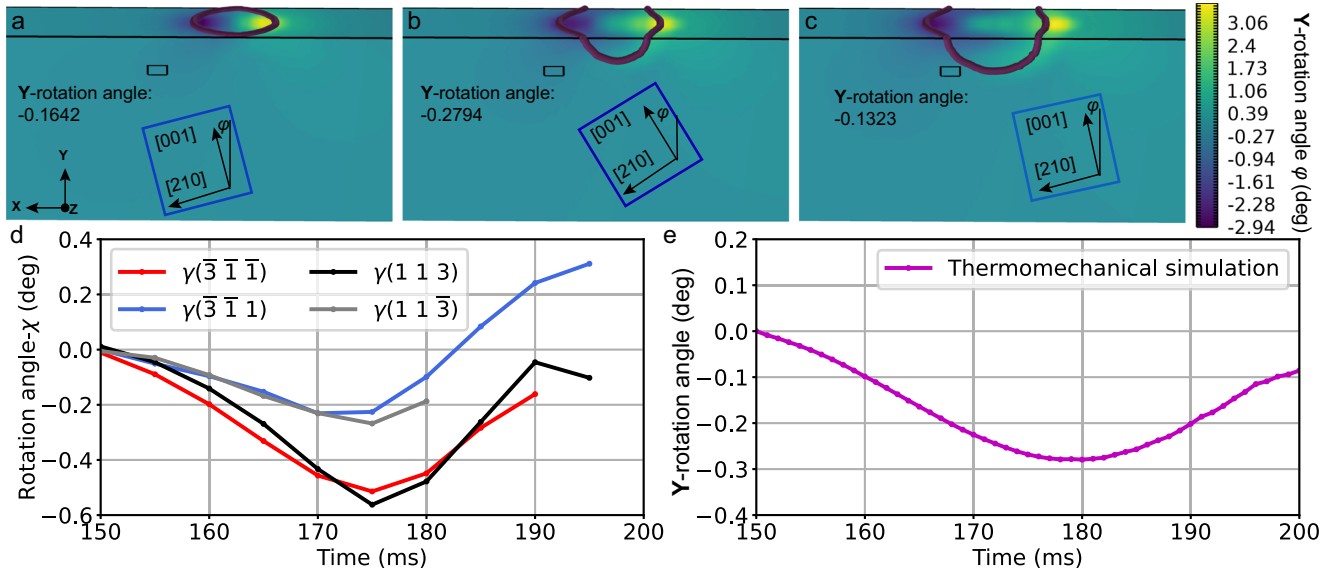

**Fig. 6 | Thermomechanical finite element simulation of the emerging Y-rotation angle during Laser Heating Periods I and II. a–c** Thermomechanical simulated **Y**-rotation angle at 165 (**a**), 180 (**b**), and 195 ms (**c**), where the mean rotation angle of the XPR and schematic crystal orientation are labelled. The melt pool is characterised by a brown tubular zone, and the cuboids represent the XPR. **d, e** Temporal evolution of the experimentally observed (**d**) and simulated (**e**) Y-rotation angles.

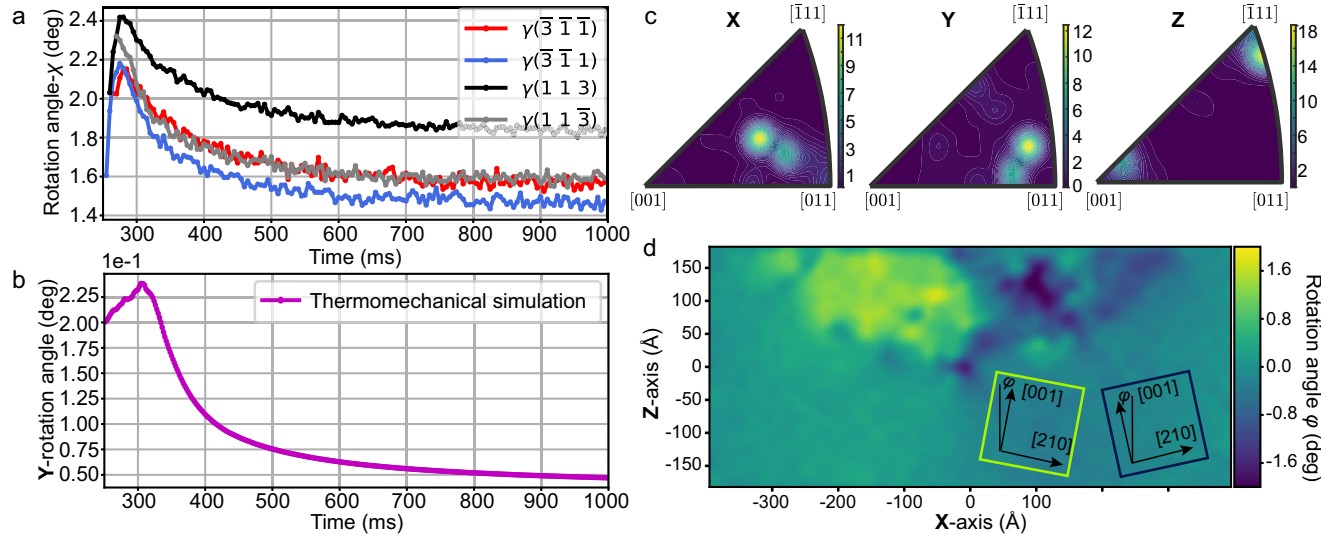

**Fig. 7 | Analysis of the grain orientation in the solidification structure. a, b** Temporal evolution of the experimental **Y**-rotation angle (**a**) and thermomechanical finite element simulated **Y**-rotation angle (**b**) during the solidification and cooling processes. **c** MD simulation of the inverse pole figure of homogeneous nucleated grains at the top of the melt pool. **d** MD simulation of the deviation of the orientation of the epitaxially grown crystal [001] at the bottom of the melt pool.

grains and released the internal stress, and then rotated[28], which explains the decreasing FWHM in Fig. 3b, d. As a consequence of the rapid movement of dislocations and nonuniform thermomechanical action, many fine grains grew into different morphologies and orientations and eventually lost their single-crystalline nature. Meanwhile, the dislocations moved to the boundary of columnar grains and gradually formed the sub-grain boundary, as illustrated in Fig. 1b, inducing the FWHM of epitaxial dendrites to be larger than that of the matrix (Fig. 3b, d). The low-intensity Laue spots of SGs imply a small grain size (Fig. 2e).

In summary, we performed in situ real-time synchrotron Laue diffraction diagnosis of the laser remelting process of nickel-based superalloys at the 3W1 beamline of BSRF and analysed the crystal rotation behaviour and stray grain formation in combination with multi-physics coupled finite element and molecular dynamics simulations. We observe significant crystal rotation along the **Y** axis during the laser remelting process and find that the localized heating heterogeneity-induced material deformation gradient field is the key driving force. Moreover, we suggest that sub-grain rotation induced by rapid dislocation movement and a complex stress field could be the main mechanism in the generation of granular SGs at the bottom of the melt pool, while the equiaxed SGs at the top of the melt pool are governed by a constitutional supercooling mechanism. The results of the Laue diffraction and EBSD show that these two types of hetero-crystal formations are closely related to laser power. Thus, reducing the introduction of dislocations and maintaining a low degree of constitutional supercooling to inhibit the formation of SGs can potentially be performed to acquire perfect additive manufactured products with a single-crystal texture. According to our findings, we expect that inhibiting crystal rotations

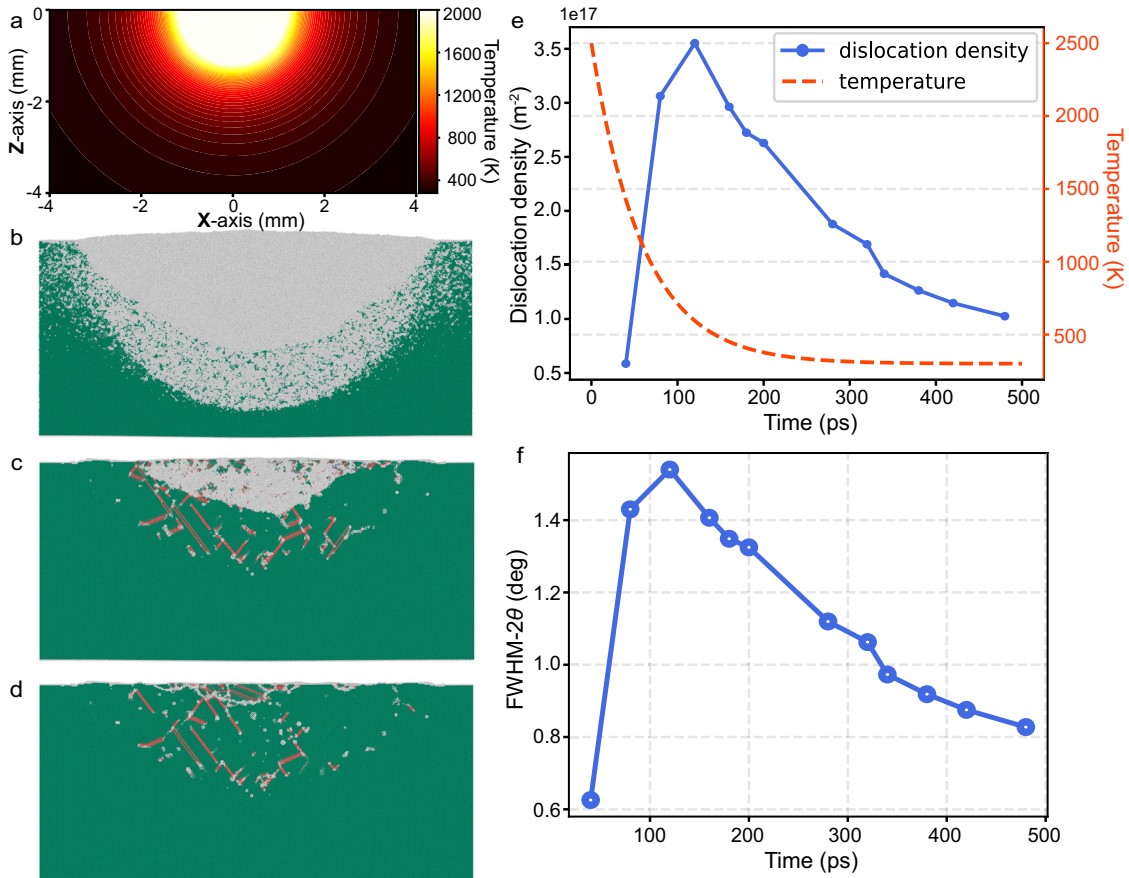

**Fig. 8 | Evolution of the defects during solidification in the MD simulation.**
**a** Spatial temperature distributions obtained from the finite element calculations.
**b** The advancement of the solid–liquid interface towards the centre of the melt pool at $t = 60$ ps. **c** The homogeneous nucleation and expansion of the grains at the centre of the melt pool at $t = 300$ ps. **d** The final solidified microstructure of the melt pool at $t = 600$ ps (where the Ni atoms coloured in green, orange, and white represent atoms in face-centered cubic (FCC), hexagonal close-packed (HCP), and amorphous environments, respectively). **e** Evolution of the dislocation density and temperature during solidification. **f** The evolution of diffraction peak broadening caused by the dislocation density as a function of time.

by decreasing the deformation gradient and maintaining a suitable temperature gradient are a possible methods to maintain the single-crystal texture along the building direction, e.g., a flat-top beam[16] and an opposite scan strategy (Supplementary Fig. 10) between interlayers would be a better choice than a Gaussian beam and an isotropic scan strategy in controlling crystal rotation during the manufacturing process.

## Methods

### Microstructure characterisation

A LEICA DMC 4500 optical microscope was adopted to characterise the microstructure of the remelted sample, which was etched by a mixture solution of 100 ml HCl + 100 ml $C_2H_2OH$ + 50 g $CuCl_2$. A JEOL JSM7100 scanning electron microscope was used to conduct electron backscattered diffraction analysis (the step size was 3 μm). The collected EBSD data were processed using TSL OIM Analysis software to calculate the **Y**-rotation of the γ phase through [100] inverse pole figures and orientation maps.

### Real-time Laue diffraction

In situ, time-resolved synchrotron Laue diffraction measurements were performed at the 3W1 beamline of the Beijing Synchrotron Radiation Facility (BSRF). The experimental setup, along with the laboratory coordinate system, is sketched in Fig. 1a. The fibre laser system produced an 800 μm diameter laser beam and propagated into the manufacturing chamber along the -**Z** direction and scanned along the **X** direction.

The transmission geometry was used for the X-ray diffraction measurements. The probe 'white' X-rays were generated from a superconducting wiggler with a broad band spectrum (Supplementary Fig. 7). The incident direction of the X-rays was -**Y**. A single-crystal tungsten pinhole was placed in front of the sample and provided a 100 μm × 100 μm square beam spot. The sample thickness was 1 mm. The EIGER X 1 M detector (Dectris Ltd) was located at 90.67 mm away from the sample downstream and operated at a frame rate of 200 Hz with the exposure time of 5 ms per frame.

### Diffraction simulation

The diffraction simulation is used to obtain the crystal orientation from the Laue diffraction patterns and the FWHM of diffraction spots induced by dislocation and micro-strain.

To exactly obtain the single crystal orientation relative to the incident X-ray, we use forward simulation of the Laue diffraction to index our X-ray diffraction patterns with the Miller indices[33,34]. Given the known information, including the experimental geometry and the "white beam" synchrotron X-ray source spectrum, we enumerate all possible orientations and calculate the corresponding X-ray diffraction patterns on the detector, which are compared with a measured diffraction pattern to find the best match[51,52].

To quantify the contribution of the strain gradient to the peak width, the strain gradient calculated from the thermomechanical simulation is input into the diffraction simulation. First, considering the size of the X-ray spot of 100 μm in the experiment, a crystal calculation area of $100 × 100 × 100$ μm³ is set, where the grid computing

cells are divided into $25 \times 25 \times 25$, each of which is set to an ideal single crystal. Then, according to the strain gradient obtained by thermomechanical simulation, the strain tensor at different positions is calculated and applied to each computing cell to deform the single crystal, based on which we can obtain the simulated diffraction pattern under the complex strain case. The contribution of the dislocation density to the peak width was calculated based on the modified Williamson-Hall method[46–48].

## Multi-physics coupled finite element modelling

The interactions between lasers and matter during additive manufacturing are complicated, and their simulations are nontrivial. Here, we use the multi-physics finite element method, implemented in COMSOL Multiphysics 6.0, to solve the 3D thermomechanical coupled problem. The 3D temperature and stress/strain fields can then be obtained.

The applied model is a weakly thermomechanical coupled model, in which the thermal field induced by laser heating is first solved. The obtained results are then inputted into the solid mechanics solver as the initial input to obtain the mechanical response of the materials[53].

In the thermal analysis, the heat transfer is governed by the time-dependent energy conservation equation, i.e., Eq. (1), where $\rho$ represents the density, $C_\rho$ is the constant pressure specific heat and $\mathbf{q}$ denotes the heat flux.

$$\begin{cases} \rho C_\rho \frac{\partial T}{\partial t} + \mathbf{q} = 0 \\ \mathbf{q} = -kT \end{cases} \tag{1}$$

The scanning laser source is modelled as a moving 2D Gaussian spot with a radius of 0.25 mm, which is treated as a boundary condition. The convective and radiative heat loss are represented by Newton's cooling law with a heat transfer coefficient of $10\,\mathrm{W\,m^{-2}\,K^{-1}}$ and the Stefan Boltzmann law with a surface emissivity of 0.7, respectively. The latent heat of the solid–liquid phase transition has also been taken into account.

In mechanical analysis, the governing equation for the stress ($\sigma$) is Eq. (2), which is derived from the conservation equations of mass and is linear. The nickel-based SX superalloy is considered an elastoplastic solid material with temperature-dependent material properties.

$$\sigma = 0 \tag{2}$$

The validation of this model, can be found in the ref. 53.

## Molecular dynamics simulations

The melting and solidification processes of nickel-based SX alloys are simulated using a large-scale atomic/molecular massively parallel simulator (LAMMPS) and embedded atom method (EAM) potential. We first construct a monocrystalline nickel-based alloy configuration with 20,000,000 atoms and an ideal face-centred cubic crystal structure. The system has dimensions of $74 \times 74 \times 35\,\mathrm{nm^3}$ along the **X, Y** and **Z** directions. Periodic boundary conditions are applied in the $X$ and $Y$ directions, and shrink-wrapped boundary conditions are applied in the **Z** direction. The sample orientation is the same as in the experiments, i.e., [211]//**X** and [001]//**Z**. The system is then relaxed for 5 ps at 300 K to reach equilibrium. The simulations are then performed in the microcanonical ensemble (NVE) with the temperature controlled by the Langevin thermostat method. The velocity-verlet algorithm is used to integrate the equations of motion with a time step of 2 fs. At different moments, the instantaneous spatial coordinates of each atom are recorded and used for structural analysis.

In MD simulations, the laser-induced heating of the material is not taken into account directly. Instead, this is realised by applying the approximate spatial-temporal thermal field derived by the multi-physics FEM results. Typical spatial temperature distributions are shown in Fig. 8a with exponentially decaying cooling rates. Within the heated area, the temperature field obtained via FEM is found to decrease approximately linearly along the radial direction and exponentially over time. These two rules are assumed to be true while generating the thermal field in MD simulations[54]. Due to the large discrepancy in the dimension between the finite element method and MD simulations, both the spatial and temporal scales in the MD simulations are accordingly reduced compared with those in the FEM simulations. Based on the temporal temperature distribution during remelting, the system is heated from 300 to 2200 K and then cooled to 300 K at different cooling rates.

## Data availability

The experimental and simulation data, as well as the simulation specifications generated in this study have been deposited in the 'Materials Cloud' database under accession code DOI: 10.24435/materialscloud:86-cq[55].

## Code availability

The multi-physics coupled FEM is implemented with the proprietary software named COMSOL Multiphysics 6.0, and the MD simulation is performed with LAMMPS, an open-source code under the GPL license. The scripts for the FEM and MD simulation in this work are displayed in the Source Data file in the 'Materials Cloud' database[55]. The X-ray diffraction simulation code for deformation is self-developed and available at https://datad.netlify.app/index.html.

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

## Acknowledgements

This research used resources from the 3W1 beamline of the Beijing Synchrotron Radiation Facility. B.Z. acknowledges the support from the National Key R&D Program of China (No. 2021YFB3703400) and the High Energy Photon Source project. W.L. acknowledges the support from the National Science and Technology Major projects 2019-VII-0003-0143 and National Natural Science Foundation (Grant No.52175369). Y.W. acknowledges the support from the National Natural Science Foundation (Grant No.52101058). S.L. acknowledges the support from National Natural Science Foundation (Grant No. 11627901). Y.T. acknowledges the support from the High Energy Photon Source project.

## Author contributions

B.Z., Y.T. and W.L. conceived the idea and coordinated the research project. B.Z., Y.T. and D.S. led the effort on developing the selected laser melting apparatus. W.L. and Y.W. prepared and characterised the sample. B.Z., D.S. and D.Z. performed the experiments with assistance from W.L. and Y.W. D.Z. processed the Laue diffraction data with help from B.Z., Y.L. and S.C. D.Z. ran the multi-physics coupled finite element model. Y.L., S.L. and S.C. contributed to the molecular dynamics and Laue diffraction simulations. All authors discussed the results. The paper was written by D.Z., B.Z., Y.L. and W.L. with inputs from all authors.

## Competing interests

The authors declare no competing interests.
