## [Peer Review File · Nature Communications]

In situ observation of crystal rotation in Ni-based superalloy during additive manufacturing processREVIEWER COMMENTS

Reviewer #1 (Remarks to the Author):

The authors applied in situ Laue diffraction to characterize the laser remelting and solidification of single-crystalline Ni superalloy samples. Thermo-mechanical and molecular dynamics (MD) simulations were performed to understand the experimental data. A new mechanism for stray grain growth was discovered – the stress induced dislocation formation and movement are responsible for stray grain formation near the interface with the substrate.

The manuscript targets an important issue in laser processing and manufacturing, that is, how to grow single-crystalline superalloy. In situ Laue diffraction was applied for the first time in this research field, while prior experiments by others all used monochromatic or pink beam for in diffraction experiments. The advantages of Laue diffraction are apparent: no need to position the single-crystalline sample in Bragg angle; multiple diffraction spots can be detected using a 2D detector simultaneously; and, a high temporal resolution can be achieved because of the high-flux incident beam. However, it's not straightforward to quantify the lattice strain while grain rotation occurs at the same time. Also, the energy bandwidth has a large contribution to the diffraction peak width.

MD modeling is a powerful tool for simulating the crystallization process, particularly when phase evolution and defect structures are of great interest. However, MD simulations require intensive computational resource, so are only performed in small length and temporal scales. This may cause discrepancies with the real solidification process involved in laser melting of metals. Therefore, MD simulations are more frequently applied in scenarios when a short-pulse laser is used.

In general, the research present here is original, which intended to use some novel approaches to address a critical issue in laser AM. The conclusion is potentially important, particularly the mechanism for the stray grain occurrence. However, the experimental and theoretical supports are not adequately strong. Therefore, I think more work are needed before it can be officially published by Nat Comm.

Below list some major concerns:

(1) Laue diffraction spots shift

In line 156 of Page 3, the authors claim “Note that the shift of Laue diffraction spots refers to the rotation of lattice planes.” The shift along θ angle usually correspond to the grain rotation, but why the authors believe the shift along 2θ is definitely caused by rotation? The authors did have lengthy discussions about “deformation field” in this manuscript, so please quantify how much the shift is due to rotation and how much is caused by lattice strain.

(2) Diffraction peak width

Many factors can contribute to the white-beam diffraction peak width, such as dislocation density (the one claimed here), gradient of lattice strain, grain size, energy bandwidth (if undulator beam), etc. The authors should first present the energy spectrum of the incident source. Then extract the thermal gradient and stress field of the crystal from the thermo-mechanical simulation. This should allow the

authors to quantify the gradient of lattice strain. The correlation between diffraction peak width and dislocation density can only be built after the devolution of the above factors.

(3) Thermo-mechanical simulation

The multi-physics finite element simulation is critical in this study because it helps explain the experiment result and also provides input for MD simulation. However, there are not many results present in the manuscript.

(4) MD simulation

The material model and process in MD simulation have the length and time scales that are orders smaller than the real solidification process of a cm-sized melt pool. The authors need to justify clearly and quantitatively that the MD simulation result is relevant here.

(5) Powder effect

The sample studied in this research was bare metal substrate without powder. The process was actually different than powder-based additive manufacturing, particularly the thermal condition. The authors need to provide strong experimental or theoretical support that the observation and conclusion apply to either powder bed or blown powder processes.

(6) Technical terms

Some of the terms used by authors are ambiguous and confusing. Please clarify the meaning of “deformation gradient field”, “rotation field”, “Brown tubular zone”, “localized heating-cooling heterogeneity”.

(7) Some other questions and comments

- What is the material composition?
- Why not use phase-field modeling to simulate the grain structures?
- Figure 5 are difficult to read and understand
- If there is crystal deformation, why the shift of diffraction spot doesn't reflect that?
- “Interestingly, according to in situ results, the dramatical decrease of FWHM during granular SGs formation period (red rectangular zone of Fig.3(b) and (d)) implies the internal relationship between the dislocations/internal stress and the formation of granular SGs.” There are a few places like this, that the conclusions were drawn before the discussions. This reduces the readability of the manuscript.

Reviewer #2 (Remarks to the Author):

The manuscripts describes an impressive experiment, time-resolved Laue diffraction, applied to an important problem, laser melt pool dynamics. The results show the dynamics of the dominant crystal grain orientation with melting and solidification and the presence of other grain (stray grains) created during solidification at the melt pool boundary.

The Laue diffraction experiments are high quality, expertly performed and well analyzed. However, the interpretations of the dynamic process created a logical argument for better characterization of the initial and final states from an experiment such as diffraction computed tomography (DCT).

For a Nature Communications paper, I recommend DCT plus the time-resolved Laue diffraction experiment. For a publication in a second tier journal, the time-resolved Laue diffraction can stand alone.

Other issues:

Typos abound: such as line 389 “grians” should be “grains”.

The video in the supplementary material uses the H264-AVF codec. As a USA scientist, this was not easily viewable. A standard Macintosh (newest operating system) could not view the view. Mathematica (newest version) could not view the video in default configuration. I was able to view the video by upgrading Mathematica with ffmpeg. That’s a lot of work to view supplementary material.

Reviewer #3 (Remarks to the Author):

Dear editor,

The manuscript aimed to study the in-situ, real-time x-ray Laue diffraction experiments to study the microstructural evolution of the second-generation nickel-based SX superalloy during the laser remelting process, capturing the dynamic crystal rotation behavior and the stray grains (SGs) formation in the melt pool. Additional thermo-mechanical coupled infinite element method and molecular dynamics simulation were employed to try to understand the solidification features.

Bellow some few comments:

Pag1 line 4 and 7 - The authors wrote, “...used in directly manufacturing of complex structure” and “...During the AM process, the metal powder...”. Not all additive manufacturing techniques can produce complex structure. The authors need to define which AM techniques they are talking about;

Pag 1 line 12 - Change k/s with K/s;

Pag 2 line 103 - Why the authors used a so low scanning speed (0.02 m/s)? For comparison is important to show the heat input of the process.

Pag 2 figure 1 – Which microscopy techniques were used to obtain the images (LOM, SEM-SE, etc.)?

Pag 3 figure 2 and pag 4 figure 3 - The $(-3-3-1)$, $(-3-31)$, (113) and $(11-3)$ diffracted spots are related with which phases? What is the experimental chemical composition of the alloy?

Dear Reviewers,

Thanks for your valuable comments on our manuscript entitled “Dynamic Crystal rocking of Nickel-Based Single Crystal Superalloy during the Epitaxial Growth of the Additive Manufacturing Process”, which greatly help us to improve the quality of this work. Substantial revision has been made to the previous version, with the point-to-point response listed below. Our response is given in normal font below this letter and the major changes/additions to the manuscript are addressed in **blue text**. The revised manuscript with major modifications **high light in yellow** is uploaded.

Response to Reviewer #1

Comment 1:

Laue diffraction spots shift

In line 156 of Page 3, the authors claim “Note that the shift of Laue diffraction spots refers to the rotation of lattice planes.” The shift along χ angle usually correspond to the grain rotation, but why the authors believe the shift along 2θ is definitely caused by rotation? The authors did have lengthy discussions about “deformation field” in this manuscript, so please quantify how much the shift is due to rotation and how much is caused by lattice strain.

Answer:

We added quantitative analysis about the influence of strain on the angular shift in the revised manuscript.

The angular shift of Laue diffraction spots in Fig. 3 could be caused by the crystal rotation and the lattice strain. We first assume that the angular shifts were independently introduced by the crystal rotation and simulate the Laue spots evolutions when the crystal rotates along the three axes, as shown in the Supplementary Movie 3, 4 and 5. One can see that the **Y**-rotation mainly corresponds to the angular shift along χ direction, while the **X** and **Z**-rotation could contribute to the shift along 2θ direction. Further comparison with the experimental data gives the relative coordinates of Euler angle of the grain orientation, at different moments during laser remelting process, as shown in Table R1.

Table R1. Euler angle of grain orientation relative coordinate during laser remelting process

	φ_X	φ_Y	φ_Z
Matrix metal at 0 ms (deg)	0.2	5	-0.5
Matrix metal at 200 ms (deg)	0.6	4.8	-0.5
Epitaxial dendrite at 300 ms (deg)	0.8	6.5	-0.5
Epitaxial dendrite at 1000 ms (deg)	0.6	6.9	-0.5

The consistence between the φ_y and the experimental data indicates the dominated rotation contribution in revealing the origin of the angular shift along χ direction. This could be further verified by the quantification of the deformation effect given by the thermo-mechanical simulation and Laue diffraction simulation¹, as shown in Fig. R1, where one can find that the strain effect only plays a minor part ($\Delta\chi=0.15 - 0.4^\circ$), in inducing the angular shift along χ direction.

Fig. R 1. Simulated rotation angle of $\gamma(\bar{3}\bar{1}\bar{1})$ (a), $\gamma(\bar{3}\bar{1}1)$ (b), $\gamma(11\bar{3})$ (c) and $\gamma(1\bar{1}\bar{3})$ (d) lattice planes along 2θ and χ directions caused by the strain during solidification by diffraction simulation.

However, just as the reviewer's comments, the ϕ_x , that is obtained under the assumption of crystal rotation, is much different with the experimental angular shift along 2θ direction in Fig. 3(c), indicates more complex mechanism of coupling of multiple factors. The thermo-mechanical assisted Laue diffraction simulation results in Fig.R1 further verified this, where the strain-induced angular shift indeed plays an important role that cannot be ignored ($0.15\sim 0.4^\circ$). Considering the relatively small scale of peak shift along 2θ direction, only the origin of \mathbf{Y} -rotation was discussed in this work.

Comment 2:

Diffraction peak width

Many factors can contribute to the white-beam diffraction peak width, such as dislocation density (the one claimed here), gradient of lattice strain, grain size, energy bandwidth (if undulator beam), etc. The authors should first present the energy spectrum of the incident source. Then extract the thermal gradient and stress field of the crystal from the thermomechanical simulation. This should allow the authors to quantify the gradient of lattice strain. The correlation between diffraction peak width and dislocation density can only be built after the devolution of the above factors.

Answer:

We added more analysis about the influence of energy bandwidth, instrument broadening, grain size, strain gradient and dislocation density on the diffraction peak width in the revised manuscript.

Energy bandwidth and instrument broadening:

As shown in Fig. R2, the probe 'white' X-rays were from a superconducting wiggler, giving a high brilliance and a broad band spectrum. For synchrotron 'white' X-rays, the diffraction peak width caused by divergence can be ignored. The broadening of detectors and other instruments remains constant in the experiment, so it will not affect the change in peak width with time.

Fig. R2. Energy spectrum of white beam X-ray generated from a superconducting wiggler

Grain size effect:

Based on the Scherrer formula, the smaller the grain size the greater the broadening and vice-versa. When the grain size reaches micron level, the peak width caused by grain size effect can be ignored and it is controlled by other factors. For additive manufactured material, as indicated by Fig. 1(b), the solidified grain size perpendicular to the build direction (**X**- and **Y**-axis) reaches micron level and that along the build direction (**Z**-axis) is bigger. Meanwhile, due to the high cooling rate, the growth of grain is quite fast². According to current temporal resolution (5ms), it is difficult to capture the process of the grain growth from nucleation to micrometer. Thus, the contribution of grain size to peak width can be ignored.

Strain gradient:

To quantify the contribution of strain gradient to peak width, the strain gradient calculated from thermomechanical simulation is inputted into diffraction simulation, according to which the strain gradient has little contribution to the diffraction peak broadening. Calculation details are shown below.

Firstly, considering the size of X-ray spot of 100 μm in the experiment, a crystal calculation area of $100 \times 100 \times 100 \mu\text{m}^3$ is set, where the grid computing cells are divided into $25 \times 25 \times 25$, each of which is set to an ideal single crystal. Then, according to the strain gradient obtained by thermomechanical simulation, the strain tensor at different positions is calculated and is applied to each computing cell to make the single crystal deformed. The Evolution of strain gradient with time during solidification is shown in Fig. R3 where the time zero point is set to the beginning of solidification.

Fig. R.3. Evolution of strain gradient with time during solidification. (a) The principal strain components at the farthest position from the center in the calculation area. (b-d) The evolution of the strain gradient tensor components associated with the principal strain with time. Time zero point is set to the beginning of solidification.

Fig. R3 (a) shows the principal strain components at the farthest position from the center in the calculation area. Since the distance is the farthest, the strain calculated from the strain gradient tensor at this point is the largest, which can reflect the evolution of the maximum strain over time. Fig. R3 (b-d) display the evolution of the strain gradient tensor components associated with the principal strain with time. It can be found that during 0 - 0.1s after solidification, the strain gradient is large. During 0.1 - 0.2 s after solidification, the strain gradient rapidly decreases. Then, part of strain gradient tensor components gradually increases in the reverse direction after about 0.2 s. Finally, all components tend to a stable value, and the length of strain gradient is smaller than the initial value.

Fig. R 4. Evolution of FWHM of $\gamma(\bar{3}\bar{1}\bar{1})$ crystal plane caused by strain gradient during solidification, where blue and orange dots represent FWHM along 2θ and χ directions, respectively. Other crystal planes show the similar results.

For this complex strain case, we performed diffraction simulation in the calculation area based on the results of Fig. R3, and obtained the evolution of half height and full width (FWHM) with time as illustrated in Fig. R4. At the beginning of solidification, the change of diffraction peak broadening is not significant. But during 0.5 – 1.5 s after solidification, the peak broadening along 2θ gradually increases while that along χ decreases. Then, the peak broadening along 2θ tends to stable while that along χ is slowly increasing. But it is noted that no matter along 2θ or χ directions, the diffraction peak width is always in a very small range ($\Delta 2\theta < 0.07^\circ$, $\Delta \chi < 0.03^\circ$) compared with our experimental results (Fig. 3). Therefore, the strain gradient has little contribution to the actual diffraction broadening.

Dislocation density:

Based on the results of MD simulation and diffraction simulation, it is found that the dramatical decrease of FWHM during SGs formation period is attributed by the evolution of dislocations. Calculation details are shown below.

Due to little contribution of other factors to peak broadening and based on the fact that the evolution trend of dislocation density (Fig. 8(e)) is consistent with that of diffraction peak broadening in Fig. 3, the correlation between diffraction peak width and dislocation density can be preliminarily established. To further verify this assumption, we conducted the diffraction simulation, where the evolution of dislocation density (Fig. 8(e)) is used as the input. And it was calculated based on the modified Williamson-Hall method³⁻⁵. The elastic constant of nickel is used in the calculation⁶. The results are plotted in Fig. R5. The diffraction simulation results about the peak broadening trend and broadening range are similar with the experimentally measurements (Fig. 3). Therefore, the dramatical decrease of FWHM during SGs formation period is due to the evolution of dislocations.

Fig. R 5. Evolution of diffraction peak broadening caused by the dislocation density with time. The nucleation just started before 100 ps.

Comment 3:

Thermo-mechanical simulation

The multi-physics finite element simulation is critical in this study because it helps explain the experiment result and also provides input for MD simulation. However, there are not many results present in the manuscript.

Answer:

In order to further understand the whole laser remelting process and better interpret the experiment data, we added more simulation results about temperature, strain and strain gradient in the revised manuscript.

The simulated temperature during laser remelting process is illustrated in Fig. R6(a), where the laser heating period III (purple rectangular zone, 200-250 ms) represents the process when the XPR is melting. At the end of laser heating period III, the temperature curve has a flat area that caused by latent heat effect, indicating the liquid-solid transition. The mean heating rate is $\sim 2.6 \times 10^4$ K/s. The mean cooling rates during 250-400 ms and 400-1000 ms are $\sim 3.4 \times 10^3$ K/s and $\sim 0.8 \times 10^3$ K/s, respectively.

The temporal evolutions of the ϵ_{XX} , ϵ_{YY} and ϵ_{ZZ} are shown in Fig. R6(b). And the representative spatial distributions of strain on substrate at 180 ms (laser heating period) and 310ms (solidification period) are displayed in Fig. R6(c-h). Obviously, strain fields are anisotropic during laser remelting process. From Fig. R6(b-e), we can find that before the material of XPR melting, as the response to the compression ϵ_{XX} and ϵ_{ZZ} fields around melt pool, the material in XPR is compressed along X and Z directions⁷, while it is stretched along Y direction. As the melt pool shifts and approaches the XPR, the compression ϵ_{XX} and ϵ_{ZZ} fields are gradually transformed into the tensile fields, which is induced by the thermal expansion. When the laser stops, the melt pool begins to solidification. The representative ϵ_{XX} , ϵ_{YY} and ϵ_{ZZ} fields during solidification are shown in Fig. R6(f-h), respectively. At the beginning of solidification, the strain is maximum, and it gradually decreases with the cooling.

Fig. R 6. Thermo-mechanical finite element simulation of emerging temperature and strain fields by laser scanning. The evolution of temperature (a) and strain (b) of XPR during laser remelting process. The ϵ_{XX} , ϵ_{YY} and ϵ_{ZZ} at 165 ms are shown in (c), (d) and (e), respectively. And that at 310 ms are displayed in (f), (g) and (h), respectively. The brown tubular zones are on behalf of the melt pool boundary and the cuboids represent the XPR.

Comment 4:*MD simulation*

The material model and process in MD simulation have the length and time scales that are orders smaller than the real solidification process of a cm-sized melt pool. The authors need to justify clearly and quantitatively that the MD simulation result is relevant here.

Answer:

Due to the limitation of computational cost, it is impossible to achieve the same length and time scales as the real additive manufacturing condition, so molecular dynamics simulation is often used as qualitative analysis of phenomena. Scale can affect the simulation results. In order to make the simulation phenomenon conform to the real situation, different simulation scales need to be tested.

In the reference [9]⁸, the influence of different scales on crystal nucleation during solidification was discussed, and it was considered that when the atomic system was large enough (>2M), the crystal nucleation phenomenon was reliable. In the paper [10]⁹, the author also carried out the molecular dynamics simulation of the additive manufacturing process, and established a system of $3438.25 \times 800 \times 57.2088 \text{ \AA}$. In our work, we have established a calculated system of $74 \times 74 \times 35 \text{ nm}^3$ containing 20000000 atoms which is large enough. At the same time, different time scales are also simulated to ensure that the time scale meets the requirements of normal crystal nucleation.

Comment 5:*Powder effect*

The sample studied in this research was bare metal substrate without powder. The process was actually different than powder-based additive manufacturing, particularly the thermal condition. The authors need to provide strong experimental or theoretical support that the observation and conclusion apply to either powder bed or blown powder processes.

Answer:

We added analysis of powder effect and provided experimental supports that the observation and conclusion apply to the existence of powder in the revised manuscript.

Experimentally, the dynamic behaviors of the additive manufacturing process with powders on it are similar with that on a bare plate, with respect to the crystal rotation behaviors observed in this work. In the comparative experiments, the X-ray was located in the interface between powder and substrate with both powder and substrate probed. The evolutions of $\gamma(11\bar{3})$ and $\gamma(113)$ diffraction spots during solidification process with and without powders are displayed in Fig. R7. Obviously, the evolution of rotation angle along both 2θ and χ directions with powders overlap well with that of the bare plate.

Fig. R 7. Dynamic evolution of $\gamma(11\bar{3})$ and $\gamma(113)$ diffraction peaks during solidification of LPBF and laser remelting process. (a-b) The rotation angle in χ (a) and 2θ (b) directions versus time. The red and blue dots represent $\gamma(11\bar{3})$ and $\gamma(113)$ diffraction peaks during LPBF, respectively. The black and green symbols represent $\gamma(11\bar{3})$ and $\gamma(113)$ diffraction peaks during laser remelting process, respectively.

Theoretically, the crystal rotation is caused by the deformation gradient which is induced by the localized heating/cooling heterogeneity upon laser scanning. The existence of the powder only leads to the slight change of heat input, which cannot affect the existence of uneven temperature field, stress field and deformation gradient that are intrinsically generated from laser scanning, so the phenomenon of crystal rotation will also occur. In conclusion, our observation and conclusion apply to the existence of powder.

Comment 6:

Technical terms

Some of the terms used by authors are ambiguous and confusing. Please clarify the meaning of “deformation gradient field”, “rotation field”, “Brown tubular zone”, “localized heating-cooling heterogeneity”.

Answer:

(1) “deformation gradient field”: Deformation gradient tensor \mathbf{F} shows how an infinitesimal line element, $d\mathbf{X}$, is mapped to the corresponding deformed line element $d\mathbf{x}$, by¹⁰

$$d\mathbf{x} = \frac{\partial \mathbf{x}}{\partial \mathbf{X}} d\mathbf{X} = \mathbf{F} d\mathbf{X}$$

\mathbf{F} contains the complete information about the local straining and the rigid rotation of the material. It is a two-point tensor, which transforms as a vector with respect to each of its indices. It involves both the reference and present configurations. In the paper, the “deformation gradient field” is used

to describe the deformation gradient of each point in space. \mathbf{F} contains the complete information about the local straining and rotation of the material. It can be decomposed by right polar decomposition into rotation tensor \mathbf{R} which describes the rigid-body rotation, and a symmetric tensor \mathbf{U} containing all information about the deformation of the material, i.e. $\mathbf{F} = \mathbf{R}\mathbf{U}$. The strain tensor $\boldsymbol{\varepsilon}$ can be calculated by \mathbf{U} . Since \mathbf{U} is independent of rigid-body rotations, this should also apply to the strain. The temporal evolution of \mathbf{R} is plotted in Supplementary Fig. 6 where the xY, yX, yZ and zY components of \mathbf{R} are almost zero during entire laser heating process, indicating that \mathbf{R} is similar to a single Y-rotation operation. Hence, the crystal Y-rotation angle χ should be mainly reflected in the xZ component of rotation tensor R_{xz} (or R_{zx}), i.e. $R_{xz} = \sin\chi$, based on which the simulated Y-rotation angle is calculated.

(2) “rotation field”: The “rotation field” is used to describe the rigid rotation angle of each point in space. To avoid misunderstanding, we modify the expression in the revised manuscript. “Obviously, spatially uneven Y-rotation field around the melt pool is present as the consequence of the localized heating heterogeneity.” is replaced by “As the consequence of the localized heating heterogeneity, the spatial distribution of the Y-rotation exhibits typical polarization characteristics, negative rotation at the front end of the melt pool and positive at the back end, as depicted in Fig. 6 (a-c).”

(3) “Brown tubular zone”: This term is misspelled, where capital letter B should be corrected to the small letter b. It was corrected in the revised manuscript. The brown tubular zones labeled in fig. F5 (a-c) are on behalf of the melt pool boundary.

(4) “localized heating-cooling heterogeneity”: This term was modified to “the localized heating/cooling heterogeneity upon laser scanning” which is used in reference [11].

Comment7:

Some other questions and comments

(1) *What is the material composition?*

(2) *Why not use phase-field modeling to simulate the grain structures?*

(3) *Figure 5 are difficult to read and understand*

(4) *If there is crystal deformation, why the shift of diffraction spot doesn't reflect that?*

(5) *“Interestingly, according to in situ results, the dramatical decrease of FWHM during granular SGs formation period (red rectangular zone of Fig.3(b) and (d)) implies the internal relationship between the dislocations/internal stress and the formation of granular SGs.” There are a few places like this, that the conclusions were drawn before the discussions. This reduces the readability of the manuscript.*

Answer:

(1) *What is the material composition?*

We added the material composition in the revised manuscript.

Chemical composition of the SX nickle-based superalloy (wt.%).

Co	Cr	Mo	W	Al	Ta	Hf	Re	Ni
7.5	7.0	1.5	5.0	6.2	6.5	0.15	3.0	Bal.

(2) *Why not use phase-field modeling to simulate the grain structures?*

Phase field (PF) model can be used to simulate the morphology evolution of complex solidification microstructures such as dendrite growth^{12,13} and segregation¹⁴. PF model is usually coupled with temperature field to simulate the evolution of solidification. Unfortunately, displacement field (or solid mechanics) which is crucial to our conclusion has not been considered into PF modeling so far. Thus, present PF modeling cannot give the evolution of defects or the explanation of crystal rotation.

(3) *Figure 5 are difficult to read and understand*

We changed the original figure 5 to figure 6 where we deleted the original Fig. 5(f) and added more descriptions about figure 6 in the revised manuscript.

As answer in comment 6 (1), the simulated crystal rotation angle is calculated based on the results of thermo-mechanical simulation. The χ -rotation angles of different areas of the substrate calculated from thermo-mechanical simulation at 165, 180 and 195 ms are shown in Fig. 6(a), (b) and (c), respectively, where the mean rotation angle of the XPR and qualitative crystal orientation are labeled. The melt pool boundary is characterized by brown tubular zone and cuboid represents the XPR. Fig. 6(d) and (e) are representative of temporal evolution of the experimentally observed and simulated χ -rotation angles of XPR during laser heating process.

(4) *If there is crystal deformation, why the shift of diffraction spot doesn't reflect that?*

As shown in Fig. 5, thermo-mechanical simulation proves the existence of crystal deformation. If there is only crystal rotation, the rotation angle of different crystal planes in χ direction should be same. As the answer in comment 1, the rotation angle of different crystal planes caused by the strain is different, which is reflected by experimental observation that all lattice planes rotate in the same direction but show subtly different rotation angle (Fig. 3a), indicating the existence of the deformation. But the contribution of deformation to peak shift is quite smaller compared with crystal rotation.

(5) *“Interestingly, according to in situ results, the dramatical decrease of FWHM during granular SGs formation period (red rectangular zone of Fig.3(b) and (d)) implies the internal relationship between the dislocations/internal stress and the formation of granular SGs.” There are a few places like this, that the conclusions were drawn before the discussions. This reduces the readability of the manuscript.*

We deleted the similar expressions in the revised manuscript, including “The evident peak broadening of diffraction spots is due to defects and inhomogeneous strain”, “Interestingly, according to in situ results, the dramatical decrease of FWHM during granular SGs formation period (red rectangular zone of Fig.3(b) and (d)) implies the internal relationship between the dislocations/internal stress and the formation of granular SGs.” and other expressions about the reason of peak broadening in Fig. 3(b) and (d) before the discussion chapter.

Response to Reviewer #2

The manuscripts describes an impressive experiment, time-resolved Laue diffraction, applied to an important problem, laser melt pool dynamics. The results show the dynamics of the dominant crystal grain orientation with melting and solidification and the presence of other grain (stray grains) created during solidification at the melt pool boundary.

Comment 1:

The Laue diffraction experiments are high quality, expertly performed and well analyzed. However, the interpretations of the dynamic process created a logical argument for better characterization of the initial and final states from an experiment such as diffraction computed tomography (DCT).

For a Nature Communications paper, I recommend DCT plus the time-resolved Laue diffraction experiment. For a publication in a second tier journal, the time-resolved Laue diffraction can stand alone.

Answer:

The DCT was not performed because there is no experimental condition. But to characterize the initial and final states, we added detailed analysis of the EBSD results in the revised manuscript as an alternative to DCT.

EBSD results of the melting pool are illustrated in Fig. R8, which reveals the crystallographic orientations in the **Z** direction (or building direction). As can be seen from Fig. R8(a), γ phase of the as-cast region of the substrate mainly exhibits one crystallographic orientation in the **Z** direction, significantly different from the one after laser melting (Fig. R8(b)). An obvious crystallographic orientation shift can be observed along the growth direction of dendrites, indicating a crystal rotation of $\sim 1.9^\circ$ between the initial and final state, along the **Y** axis (**Y**-rotation). Apparent crystal rotation relative to the initial substrate was also verified by our in-situ Laue diffraction measured results, with almost the same rotation angle.

Fig. R 8. EBSD results of substrate (a) and final state (b) after laser remelting process, where [100] direction is along **Z** direction. In (b), the blue and green colors indicate (1 0 30) and (0 0 1) crystal

plane along Z direction, respectively.

Comment 2:

Typos abound: such as line 389 “grians” should be “grains”.

Answer:

We changed “grians” with “grains” in the revised manuscript.

Comment 3:

The video in the supplementary material uses the H264-AVF codec. As a USA scientist, this was not easily viewable. A standard Macintosh (newest operating system) could not view the view. Mathematica (newest version) could not view the video in default configuration. I was able to view the video by upgrading Mathematica with ffmpeg. That’s a lot of work to view supplementary material.

Answer:

We changed the video format to H.264 encoding according to the requirements of the journal which should be easily viewable.

Response to Reviewer #3

Dear editor,

The manuscript aimed to study the in-situ, real-time x-ray Laue diffraction experiments to study the microstructural evolution of the second-generation nickel-based SX superalloy during the laser remelting process, capturing the dynamic crystal rotation behavior and the stray grains (SGs) formation in the melt pool. Additional thermo-mechanical coupled infinite element method and molecular dynamics simulation were employed to try to understand the solidification features.

Comment 1:

Pag1 line 4 and 7 - The authors wrote, "...used in directly manufacturing of complex structure" and "...During the AM process, the metal powder...". Not all additive manufacturing techniques can produce complex structure. The authors need to define which AM techniques they are talking about;

Answer:

We modified the expression in the revised manuscript.

Additive manufacturing (AM) such as laser selective melting (SLM) and direct energy deposition (DED) is driven by three-dimensional (3D) digital model of the parts and used in directly manufacturing of complex structure without any mold, which has been considered as a revolutionary breakthrough in the field of manufacturing technology.

During the LSM or DED process, the metal powder would be melted instantly and a microscale molten pool forms in a short time under the operation of a high-energy beam such as a laser, achieving steep temperature gradients of up to $\sim 10^7$ K/m and ultra-high cooling rates of $\sim 10^7$ K/s.

Comment 2:

Pag 1 line 12 - Change k/s with K/s;

Answer:

We changed "k/s" with "K s⁻¹" in the revised manuscript.

Comment 3:

Pag 2 line 103 - Why the authors used a so low scanning speed (0.02 m/s)? For comparison is important to show the heat input of the process.

Answer:

Due to present experimental condition that the flux of X-ray is not high enough and the detection efficiency of low-energy detectors (EIGER X 1 M) for high-energy photons is relatively low, only 5ms temporal resolution can be achieved. To capture the behavior of nickel-based SX superalloy before melting and during solidification, the scanning speed should be low. Otherwise, the crystal rotation behavior during laser heating as illustrated in Fig. 6(b) cannot be observed, because laser would soon reach the X-ray probed area and melt the material.

We added the input energy density in the revised manuscript.

Input energy density (IED) is defined as laser power, P, over the product of laser scan speed, v, and

the laser beam diameter, d , i.e. $IED = P/(v \cdot d)$ with a unit of $J m^{-2}$.¹⁵ The laser power, scan speed and laser beam diameter are 266 W, $0.02 m s^{-1}$ and $800 \mu m$, respectively. As a result, the IED is $1662.5 J cm^{-2}$.

Comment 4:

Pag 2 figure 1 – Which microscopy techniques were used to obtain the images (LOM, SEM-SE, etc.)?

Answer:

It was obtained through optical microscope. We added a new sub-chapter ‘Microstructure characterization’ in the ‘Method’ chapter.

A LEICA DMC 4500 optical microscope was adopted to characterize microstructure of the remelted sample, which was etched by a mixture solution of 100 ml HCl + 100 ml C_2H_2OH + 50 g $CuCl_2$. A JEOL JSM7100 scanning electron microscope was used to conduct electron backscattered diffraction analysis (step size was $3 \mu m$). The collected EBSD data were processed using TSL OIM Analysis software to calculate Y-rotation of γ phase through [100] inverse pole figures and orientation maps.

Comment 5:

Pag 3 figure 2 and pag 4 figure 3 - The (-3-3-1), (-3-31), (113) and (11-3) diffracted spots are related with which phases? What is the experimental chemical composition of the alloy?

Answer:

The diffracted spots in figure 2 and figure 3 are γ phase in nickle-based superalloy.

We added the material composition in the revised manuscript.

Chemical composition of the SX nickle-based superalloy (wt.%).

Co	Cr	Mo	W	Al	Ta	Hf	Re	Ni
7.5	7.0	1.5	5.0	6.2	6.5	0.15	3.0	Bal.

References

1. Huang, J. W., Zhang, Y. Y., Hu, S. C., Cai, Y. & Luo, S. N. DATAD: A Python-based X-ray diffraction simulation code for arbitrary texture and arbitrary deformation. *J Appl Crystallogr* **54**, 686–696 (2021).
2. Gao, Z. & Ojo, O. A. Modeling analysis of hybrid laser-arc welding of single-crystal nickel-base superalloys. *Acta Mater* **60**, 3153–3167 (2012).
3. Ungár, T. & Borbély, A. The effect of dislocation contrast on x-ray line broadening: A new approach to line profile analysis. *Appl Phys Lett* **69**, 3173–3175 (1996).
4. Unga, T., Dragomir, I., Re, A. & Ve, S. *DISLOCATIONS, GRAIN SIZE AND PLANAR FAULTS IN NANOSTRUCTURED COPPER DETERMINED BY HIGH RESOLUTION X-RAY DIFFRACTION AND A NEW PROCEDURE OF PEAK PROFILE ANALYSIS*.
5. Unga, T., Dragomir, I., Re, A. & Ve, S. & Borbe, A. *The contrast factors of dislocations in cubic crystals: the dislocation model of strain anisotropy in practice*. *J. Appl. Cryst* vol. 32 (1999).
6. Neighbours, J. R., Bratten, F. W. & Smith, C. S. The Elastic Constants of Nickel. *J Appl Phys* **23**, 389–393 (1952).
7. Calta, N. P. *et al.* Cooling dynamics of two titanium alloys during laser powder bed fusion probed with in situ X-ray imaging and diffraction. *Mater Des* **195**, (2020).
8. Mahata, A. & Zaeem, M. A. Size effect in molecular dynamics simulation of nucleation process during solidification of pure metals: investigating modified embedded atom method interatomic potentials (2019 Modelling Simul. Mater. Sci. Eng. 27 085015). *Model Simul Mat Sci Eng* **28**, 019601 (2020).
9. Kurian, S. & Mirzaeifar, R. Selective laser melting of aluminum nano-powder particles, a molecular dynamics study. *Addit Manuf* **35**, 101272 (2020).
10. Richard B. Hetnarski & M. Reza Eslami. *Thermal Stresses—Advanced Theory and Applications*. vol. 158 (Springer Cham, 2019).
11. Wang, G. *et al.* The origin of high-density dislocations in additively manufactured metals. *Mater Res Lett* **8**, 283–290 (2020).
12. Körner, C., Markl, M. & Koepf, J. A. Modeling and Simulation of Microstructure Evolution for Additive Manufacturing of Metals: A Critical Review. *Metallurgical and Materials Transactions A* **51**, 4970–4983 (2020).
13. Wang, L., Wei, Y., Zhan, X. & Yu, F. A phase field investigation of dendrite morphology and solute distributions under transient conditions in an Al–Cu welding molten pool. *Science and Technology of Welding and Joining* **21**, 446–451 (2016).
14. Nomoto, S., Kusano, M., Kitano, H. & Watanabe, M. Multi-Phase Field Method for Solidification Microstructure Evolution for a Ni-Based Alloy in Wire Arc Additive Manufacturing. *Metals (Basel)* **12**, (2022).
15. Guo, Q. *et al.* In-situ characterization and quantification of melt pool variation under constant input energy density in laser powder bed fusion additive manufacturing process. *Addit Manuf* **28**, 600–609 (2019).

REVIEWERS' COMMENTS

Reviewer #1 (Remarks to the Author):

My comments on the original manuscript were all well addressed. The authors' efforts on performing more simulations and data analyses are highly appreciated. This work sets a good example of using in situ synchrotron diffraction techniques to study laser additive manufacturing process.

Reviewer #2 (Remarks to the Author):

This paper does not read well. The time resolved Laue data as a function of laser power is valuable data. However, the presentation and interpretation of the data lacks clarity. Similar data presentations are given in references 14 Chauvet (2018) and 16 Jodi (2022). The Chauvet and Jodi papers have clarity and organization which the present paper should aspire to match.

Supp Fig 1 and the 266 W movie are good starting points. The authors then generate Fig 2 and 3 fairly nicely. However, the Supp movies for explaining Laue spots as a function of rotation about axes are useless as the filenames are lost; the axes labels must be include as annotations within the movies.

At Fig 4, the paper moves beyond the supporting data. Fig 4a lacks labeling to indicate the time axes is the same Fig 4a as in Fig 3a. Second, where is the raw data? Supp movie 280 W might be a partial presentation of the raw data. Supp Fig 3c,d do not look to be in agreement with Fig 4b at the 350 W and 385 W power levels.

I have no idea how Table 1 is create. There is a tiny axes triad in Fig 1a. The lack of detail mean the raw data could not be reevaluated by other researchers to arrive at Table 1. The work is not reproducible.

Page 16: Stray grains. The discussion appears rushed and is difficult to follow. Yet, the topic is a key component of the abstract.

Writing: There are strange sentences throughout. Page 2. "However, previous researches on the mechanisms of microstructural evolution during the epitaxial growth process are mainly revealed by the reverse deduction of the recovery specimens and numerical simulation".

Reviewer #3 (Remarks to the Author):

The authors answered all questions, and now the manuscript can be published.

Dear Reviewers,

Thanks for your valuable comments on our manuscript entitled “Dynamic Crystal rocking of Nickel-Based Single Crystal Superalloy during the Epitaxial Growth of the Additive Manufacturing Process”, which greatly help us to improve the quality of this work. Substantial revision has been made to the previous version, with the point-to-point response listed below. Our response is given in normal font below this letter and the major changes/additions to the manuscript are addressed in blue text. The revised manuscript with major modifications high light in yellow is uploaded.

Response to Reviewer #2

Comment 1:

This paper does not read well. The time resolved Laue data as a function of laser power is valuable data. However, the presentation and interpretation of the data lacks clarity. Similar data presentations are given in references 14 Chauvet (2018) and 16 Jodi (2022). The Chauvet and Jodi papers have clarity and organization which the present paper should aspire to match.

Answer:

Chauvet¹ and Jodi² papers show great organization and better readability. They present the ex-situ microstructure characterization results, and then followed by discussions. In our article, we spend lengthy paragraphs to describe two different kinds of experimental phenomenon, crystal rotation and the SGs generation, and then discuss their mechanism in a separated section, which will reduce the readability of the article. We modify several paragraphs and sentences of the manuscript to make it more organized. The description and interpretation of Fig. 4 and Table 1 in the last version are insufficient. To address this issue, we have provided additional details and interpretation of Fig. 4 in the revised manuscript. Additionally, in response to Comment 4, we have included the calculation method for Table 1.

Comment 2:

Supp Fig 1 and the 266 W movie are good starting points. The authors then generate Fig 2 and 3 fairly nicely. However, the Supp movies for explaining Laue spots as a function of rotation about axes are useless as the filenames are lost; the axes labels must be included as annotations within the movies.

Answer:

Thanks for the tip. we are sorry for the miss of the annotations in the supplementary movies, which will surely confuse the readers. The axes of rotation are labeled in the supplementary movies, while annotations regarding the presence or absence of powder are also included. Supplementary Movie 6: Simulated shift of diffraction spots when the crystal rotates along the **X**-axis; Supplementary Movie 7: Simulated shift of diffraction spots when the crystal rotates along the **Y**-axis; Supplementary Movie 8: Simulated shift of diffraction spots when the crystal rotates along the **Z**-axis

Comment 3:

- (1) At Fig 4, the paper moves beyond the supporting data. Fig 4a lacks labeling to indicate the time axes is the same Fig 4a as in Fig 3a. Second, where is the raw data? Supp move 280W might be a partial presentation of the raw data.
- (2) Supp Fig 3c,d do not look to be in agreement with Fig 4b at the 350 W and 385 W power levels.

Answer:

(1) Both Fig. 4a and Fig. 3 were generated using the same raw data obtained at a laser power of 266 W. In the original manuscript, only a portion of the complete experimental data was cut off and presented in Fig. 3a, in order to clearer illustrate the angular shift between 150-250 ms.. In the revised manuscript, Fig. 3a was redrawn to align with the time axis of Fig. 4.

In the new version, we included more data in the supplementary material, including Supplementary movies depicting the evolution of Laue diffraction images at different laser powers of 245W, 350W, and 385W. These diffraction peaks exhibit behavior similar to those at the laser of 266 W. e.g. Fig. R1 presents the rotation angle and full width at half maximum (FWHM) in the χ and 2θ directions over time at a laser power of 245W, which is similar to Fig. 3 of the manuscript.

Fig. R1 Dynamic evolution of diffraction peaks during laser remelting process under 245W. (a-b) The rotation angle (a) and FWHM (b) in the χ direction versus time. (c-d) The rotation angle (c) and FWHM (d) in the 2θ direction versus time. The red, blue, black and green dots represent $\gamma(\bar{3}\bar{1}\bar{1})$, $\gamma(\bar{3}\bar{1}1)$, $\gamma(11\bar{3})$ and $\gamma(11\bar{3})$ lattice planes, respectively. The orange, green, purple and red rectangular zones denote four different stages of the laser remelting process. The laser was switched on at 150 ms and off at 250 ms.

Supplementary movie 280W depicts the evolution of diffraction patterns during the laser powder bed fusion process, not the laser remelting process. The supplementary movies also contain annotations indicating the presence or absence of powder. We compared the behavior of diffraction spots during the laser powder bed fusion process under different power levels and observed similar crystal rotation behaviors (Supplementary Fig. 5) and the effect of laser power on the number of

spots (Supplementary Fig. 4).

(2) The number of diffraction spots is used to qualitatively characterize the fraction of SGs. Here, we use a relative intensity threshold when comparing among different laser powers, but not the absolute intensity threshold. This is because the intensity of single-crystal diffraction spots decreases with the increase of laser power, as demonstrated in Fig. R2. Then an intensity threshold of 0.15-times the strongest intensity from single-crystal diffraction spots is set to estimate the SGs fraction of different samples instead of a constant intensity threshold. Therefore, although Support Fig 3d looks like having more diffraction spot but some of them are below the threshold and haven't been counted in the fraction of SGs.

Fig. R2 The intensity of four single-crystal diffraction spots as a function of laser power.

Then, we investigated the effect of different intensity thresholds (0.10-times, 0.15-times, and 0.20-times the strongest intensity) on the extraction of diffraction spots, as shown in Fig. R3. Although the number of diffraction spots varies with the three different thresholds, they all show the same trend. Additionally, the results obtained from samples with and without powder (Fig. R3 and Supplementary Fig. 4) are similar.

Fig. R3 Number of diffraction spots as a function of laser power under different threshold: (a) 0.10-times, (b) 0.15-times and (c) 0.15-times the strongest intensity.

Compared to Supplementary Fig. 3(d), Supplementary Fig. 3(c) has more low-intensity spots which are below the set intensity threshold. Therefore, visually, Supplementary Fig. 3(c) and (d) do not seem to agree with Fig. 4(b), especially after adjusting the image contrast. To avoid ambiguity, we replaced Supplementary Fig. 3(c) with another dataset that has the same experimental parameters as Supplementary Fig. 3(c). The new results are presented in Fig. R4.

Fig. R4 Final-state Laue diffraction patterns captured after solidification under different laser power. Four laser powers are used: 245 W (a), 266 W (b), 350 W (c) and 385 W (d).

Comment 4:

I have no idea how Table 1 is created. There is a tiny axes triad in Fig 1a. The lack of detail mean the raw data could not be reevaluated by other researchers to arrive at Table 1. The work is not reproducible.

Answer:

We added detailed calculation method about Table 1 in ‘Methods’ section.

To exactly obtain the single crystal orientation relative to the incident X-ray, we use forward simulation of the Laue diffraction to index our X-ray diffraction patterns with the Miller indices^{3,4}. Given the known information, including the experimental geometry and the “white beam” synchrotron X-ray source spectrum, we enumerate all possible orientations and calculate the corresponding X-ray diffraction patterns on the detector, which are compared with a measured diffraction pattern to find the best match^{5,6}. Here, we follow these procedures to perform forward simulation to index our diffraction pattern:

1. Analyse data: We extract the positions, presented by 2θ and χ pairs, and the intensities of

diffraction spots on the detector based on actual calibrated experimental geometry. Actual values of positions and intensities are obtained by 2D Gaussian fitting in the areas surrounding the spots.

2. Generate candidates: An initial single crystal in the experimental coordinate systems with the desired orientation as the initial orientation is constructed. Then, we enumerate rotations whose misorientations are in the range of a predefined maximum angle with certain precision and apply the rotations to the initial single crystal to generate candidate single crystals.

3. Simulated diffraction: For each alternative single crystal, the 2θ and χ pairs and intensities of the diffraction spots with Miller indices less than the predefined maximum Miller indices are calculated based on the incident direction of X-rays in the experimental coordinate systems. Here, the spots with zero intensities caused by structure extinction are filtered.

4. Search: First, we search for those candidates whose positions of the diffraction spots coincide with the positions from experimental data in a predefined bias. The most consistent indices of the experimental diffraction spots and single crystal orientation may be determined after the above search.

Comment 5:

Page 16: Stray grains. The discussion appears rushed and is difficult to follow. Yet, the topic is a key component of the abstract.

Answer:

We modified this section to make it more logical and easier to follow.

Comment 6:

Writing: There are strange sentences throughout. Page 2. "However, previous researches on the mechanisms of microstructural evolution during the epitaxial growth process are mainly revealed by the reverse deduction of the recovery specimens and numerical simulation".

Answer:

We performed the English language editing service, recommended by the editorial office

Reference:

1. Chauvet, E., Tassin, C., Blandin, J.-J., Dendievel, R. & Martin, G. Producing Ni-base superalloys single crystal by selective electron beam melting. *Scr Mater* **152**, 15–19 (2018).
2. Jodi, D. E., Kitashima, T., Koizumi, Y., Nakano, T. & Watanabe, M. Manufacturing single crystals of pure nickel via selective laser melting with a flat-top laser beam. *Additive Manufacturing Letters* **3**, 100066 (2022).
3. Huang, X. R. LauePt, a graphical-user-interface program for simulating and analyzing white-beam X-ray diffraction Laue patterns. *J Appl Crystallogr* **43**, 926–928 (2010).
4. Huang, J. W., Zhang, Y. Y., Hu, S. C., Cai, Y. & Luo, S. N. DATAD: A Python-based X-ray diffraction simulation code for arbitrary texture and arbitrary deformation. *J Appl Crystallogr* **54**, 686–696 (2021).
5. Zhang, Y. Y. *et al.* Ultrafast X-Ray Diffraction Visualization of B1-B2 Phase Transition

- in KCl under Shock Compression. *Phys Rev Lett* **127**, (2021).
6. Gupta, V. K. & Agnew, S. R. Indexation and misorientation analysis of low-quality Laue diffraction patterns. *J Appl Crystallogr* **42**, 116–124 (2009).